# Signals, Concepts, and Laws: Toward Universal, Explainable Time-Series Forecasting

## Abstract

Accurate, explainable and physically credible forecasting remains a persistent challenge for multivariate time-series with domain-varying statistical properties. We propose DORIC, a Domain-Universal, ODE-Regularized, Interpretable-Concept Transformer for Time-Series Forecasting that generates predictions through five self-supervised, domain-agnostic concepts while enforcing differentiable residuals grounded in first-principles constraints. The concepts are softly regressed toward analytic statistics of the raw signal, and a driven–damped ODE head couples these concepts to the forecast as a shared, mean-reverting dynamical template across datasets. Unlike prior efficiency-focused Transformers, such as Informer(sparse attention) or FEDformer(frequency priors), DORIC combines latent explainability with explicit scientific constraints, while preserving the attention mechanism's capacity to model long-range dependencies. We evaluate DORIC on six publicly-available datasets and it achieves the lowest error in eight of twelve MSE/MAE metrics. Compared with TimeMixer, DORIC outperforms it on four datasets while maintaining strong interpretability. Interpretability analyses show that the learned concepts remain strongly aligned with their analytic targets, physics residuals stay relative to the signal scale, and the learned ODE coefficients follow domain-consistent patterns. Ablation studies reveal complementary contributions: removing the physics residual increases average MSE from 0.328 to 0.547, eliminating concept alignment raises it to 0.698, and replacing the shared encoder with disjoint concept heads results in a 76% increase.

## 1 Introduction

Real-world decision systems, from power-grid dispatch and urban mobility control to epidemic surveillance and high-frequency trading, depend on fast, accurate, and trustworthy forecasts of multivariate time-series. Classical linear models, exemplified by the Box-Jenkins ARIMA methodology (Tunnicliffe Wilson, 2016), provide statistical rigor but struggle with high-frequency, non-stationary data or dynamics governed by latent physical constraints. Over the past decade, attention-based deep learning has revolutionized sequential modelling: the Transformer architecture introduced by Vaswani et al. (2017) replaced recurrence with self-attention, enabling parallel learning over long-contexts. Subsequent efficiency-oriented variants, such as Reformer (Kitaev et al., 2020) and LogTrans (Li et al., 2019), reduce memory overhead via locality-sensitive hashing and log-sparse kernels. Meanwhile, domain-specific advances, such as Informer (Zhou et al., 2021), Autoformer (Wu et al., 2021), and FEDformer (Zhou et al., 2022), introduced decomposition, auto-correlation and frequency sparsity to enhance long-horizon accuracy. More recently, PatchTST (Nie et al., 2023b) demonstrated further gains via patching and exponential-smoothing priors. A comprehensive survey (Wen et al., 2023) attests to the field's explosive growth. Despite these advancements, three structural gaps remain.

- **Gap 1 – Physics non-compliance.** Purely data-driven Transformers can produce trajectories that violate fundamental constraints (e.g., mass-balance or energy conservation), thereby undermining stakeholder trust and limiting deployment in safety-critical domains.
- **Gap 2 – Latent opacity.** Although post-hoc explanation methods exist, their fidelity degrades in multivariate settings. In contrast, concept-driven transparency pioneered by Concept Bottleneck Models (Koh et al., 2020) and their stochastic extensions (Vandenhirtz et al., 2024) remains under-explored in the context of time-series forecasting.

- **Gap 3 – Lack of universality.** Most specialised architectures are tailored to specific domains (e.g., weather or finance) and struggle to generalise across varying sampling rates, noise characteristics, and seasonality patterns.

To bridge these gaps, we view long-horizon forecasting through an *interpretability-first* lens. Rather than adding ever more opaque modules, we ask whether a single concept–physics layer can serve as a reusable interface between diverse multivariate time series and human experts. Concretely, we introduce **DORIC**, a forecasting architecture that augments a standard encoder with (i) a small set of physically motivated concepts computed from the input signals, and (ii) a physics-informed head that couples these concepts via simple laws. This yields a model whose internal "dials" have stable semantics across datasets while remaining competitive with strong baselines.

**Contributions.** The contributions of this work are threefold:

- **A universal concept–physics layer for multivariate time series.** We design five low-level yet broadly applicable concepts (level, velocity, instantaneous power, dominant periodic amplitude, and local volatility) that can be attached to standard sequence encoders. These concepts are computed by causal statistics and serve as a shared vocabulary across datasets, rather than being tailored to any single benchmark.
- **A physics-informed head with provable training dynamics.** We couple the concepts to the forecast through a driven–damped first-order ODE and turn this ODE into algebraic residuals. The resulting physics loss admits a ramp-up schedule under which SGD converges while the physics violation vanishes. This connects the training trajectory to a clear three-stage story: feasibility (physics residual collapse), concept alignment, and data fit.
- **An interpretability-first evaluation of forecasting performance.** We show on seven standard benchmarks that plugging the concept–physics layer into a strong Transformer backbone achieves accuracy comparable to recent state-of-the-art models, while enabling concept-level analyses, gradient-based local sensitivities, and cross-dataset comparisons. Ablations demonstrate how concept supervision and physics penalties jointly control the trade-off between predictive accuracy and interpretability.

**Scope of interpretability and scientific value.** Our interpretability claim is scoped at the level of the shared bottleneck and dynamical template rather than at every individual weight in the network. Each forecast $\hat{y}_t$ is required to pass through a five-dimensional concept vector $c_t = (c_{1,t}, \ldots, c_{5,t})$ and a driven–damped ODE head that are shared across all datasets. These coordinates are not arbitrary hidden units: they are softly regressed towards analytic statistics of the raw signal (sliding mean, local velocity, instantaneous power, dominant periodic amplitude, local volatility), so that the bottleneck remains tied to physically meaningful quantities instead of drifting into opaque features. The ODE head then combines these concepts through a mean-reverting dynamics that is also shared across domains. Together, this design yields a mediating mechanism from past observations to future predictions: time points influence the forecast only through how they shape the concept trajectories and how those trajectories are processed by the ODE. We do not claim that these five concepts exhaust all possible explanations, but that DORIC enforces a low-dimensional, physically anchored interface that is consistent across heterogeneous time-series domains.

## 2 LITERATURE REVIEW

Early time-series forecasting relies on stochastic linear models such as the Box–Jenkins ARIMA (Tunnicliffe Wilson, 2016), whose identification–estimation–diagnosis cycle remains influential but often fails under the non-stationarity of modern telemetry.

Deep learning models capture time-series patterns with specifically-designed architectures spanning a wide range of foundation backbones, including CNNs (Wang et al., 2023; Wu et al., 2023; Hewage et al., 2020), RNNs (Lai et al., 2018; Qin et al., 2017; Salinas et al., 2020), Transformers (Vaswani et al., 2017) and MLPs (Zeng et al., 2023b; Zhang et al., 2022; Oreshkin et al., 2019; Challu et al., 2023). The quadratic cost of vanilla attention spurred efficient variants, Reformer (Kitaev et al., 2020) and LogTrans (Li et al., 2019), and domain-specific advances: Informer's ProbSparse attention (Zhou et al., 2021), Autoformer's trend–seasonality decomposition (Wu et al., 2021), FEDformer's Fourier sparsity (Zhou et al., 2022), PatchTST's patch-token strategy (Nie et al., 2023b),

and TimeMixer (Wang et al., 2024). A 2023 survey (Wen et al., 2023) documents more than 120 Transformer adaptations for time-series forecasting.

Despite these advances, interpretability remains a challenge, prompting growing interest in concept-level supervision. Concept Bottleneck Models (Koh et al., 2020) demonstrate that forcing predictions through human-meaningful variables can maintain accuracy while enabling user interventions. Stochastic CBMs (Vandenhirtz et al., 2024) generalized this idea by modelling concept dependencies probabilistically. FF Bottleneck (van Sprang et al., 2025) used an AR model as a surrogate concept within a Transformer architecture; however, empirical results revealed that the standalone AR model outperformed FF Bottleneck on four out of the six datasets.

In parallel, physics-informed learning has emerged as a remedy for physically implausible outputs. Raissi's PINNs framework (Raissi et al., 2019) pioneered the joint optimization of data and governing equations; PGTransNet (Wu et al., 2024) integrated physics-guided self-attention for Pacific-Ocean temperature forecasting, while physics-informed LSTM variants have been applied to power-transformer health monitoring (Lei et al., 2025). The PhysicsSolver (Zhu et al., 2025) further attests to the potential of incorporating scientific priors into deep learning architectures.

Hybrid work now marries physics with deep networks: a TCN–TFT ensemble improves probabilistic wind forecasts (Mi et al., 2025), while energy-constrained diffusion Transformers extend long-horizon accuracy (Ren et al., 2024). Interpretability has advanced through Temporal Fusion Transformer's gating and variable-selection (Lim et al., 2021) and through sparse-attention schemes such as Adversarial Sparse Transformer (Wu et al., 2020) and Query-Selector attention (Klimek et al., 2022). Complementing advances in model architecture, evaluation paradigms have matured: GraphCast — a graph-neural extension that outperforms ECMWF simulations on 90% of medium-range weather targets (Lam et al., 2023) —demonstrates that ML systems can rival numerical solvers at global scale. Meanwhile, foundation models like TimeGPT (Garza et al., 2024) deliver strong zero-shot generalization capabilities across diverse time-series domains. Nevertheless, critical evaluations have shown that simple linear baselines can outperform many Transformer variants when evaluated rigorously, as highlighted in the "Are Transformers Effective for Time-Series Forecasting?" paper (Zeng et al., 2023a). Additionally, the rise of diffusion-based Transformers (Fu et al., 2025) reflects the field's continued pursuit of the right balance between model complexity and effective inductive biases.

In summary, although current approaches span efficiency improvements, frequency-domain finesse, multiresolution structure and physics-guided regularization, no existing model simultaneously guarantees physical plausibility, concept-level transparency and cross-domain universality. DORIC fills this gap by fusing a novel five-concept bottleneck with analytic residual constraints inside a Transformer backbone and validating its effectiveness across six heterogeneous benchmarks.

## 3 METHODOLOGY

This section introduces the methodology of DORIC. The overall architecture of DORIC is illustrated in Figure 1, time series data first enters the transformer encoder, then undergoes conceptual layer transformation and physical layer constraints, and finally generates a prediction..

### 3.1 RAW DATA AND FORECASTING GOAL

We start from a single-channel time series

$$\mathbf{y}_{1:T} = (y_1, y_2, \ldots, y_T) \in \mathbb{R}^T \tag{1}$$

where $T \in \mathbb{N}$ denotes the experiment length (e.g., $T = 24\,000$ for three years of hourly data), and $y_t$ is the raw observation at discrete time index $t$ in its original physical unit.

At forecast step $t\,(> L)$ we reveal the strictly causal window

$$\mathbf{y}_{t-L:t-1} = (y_{t-L}, \ldots, y_{t-1}) \in \mathbb{R}^L, \quad L \geq \tau + 1, \tag{2}$$

where $L$ is the look-back horizon (e.g. $L = 120$ for five days for hourly or five months for daily-weekly signals), and $\tau$ is the conceptual sub-window used later in equation (10), e.g. $\tau = 50$.

Our forecasting operator is therefore a map

$$f_\Theta : \mathbb{R}^L \longrightarrow \mathbb{R}, \qquad \hat{y}_t = f_\Theta(\mathbf{y}_{t-L:t-1}), \tag{3}$$

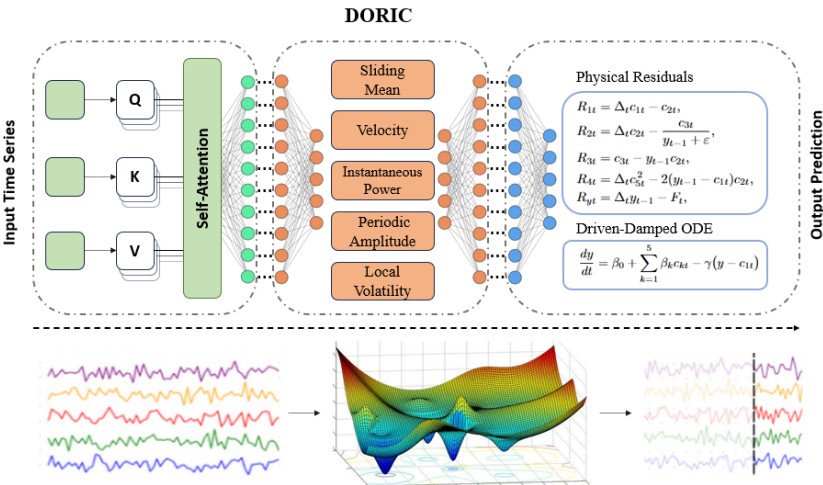

Figure 1: DORIC Structure

where $\Theta$ collects all trainable parameters of the network (weights, biases, and physics coefficients).

### 3.2 THE CAUSAL TRANSFORMER ENCODER

The encoder converts the raw window, denoted by $\mathbf{y}_{t-L:t-1}$, into a dense vector $\mathbf{z}_t \in \mathbb{R}^d$, where $d$ is the dimension of the latent embedding, summarizing all past information. The encoder consists of the following layers:

#### 3.2.1 INITIAL TOKEN EMBEDDING:

Let

$$\mathbf{H}^{(0)} = \underbrace{\mathbf{y}_{t-L:t-1}\mathbf{W}_e}_{\text{value} \to \text{vector}} + \underbrace{\mathbf{P}}_{\text{position}}, \tag{4}$$

where $\mathbf{W}_e \in \mathbb{R}^{1 \times d}$ (row) is the embedding matrix, mapping each scalar value to a $d$ dimensional vector, $\mathbf{P} \in \mathbb{R}^{L \times d}$ encodes absolute positions via sinusoidal basis: $P_{i,2k} = \sin(i/10^{4k/d})$, $P_{i,2k+1} = \cos(i/10^{4k/d})$, and $\mathbf{H}^{(0)} \in \mathbb{R}^{L \times d}$ is the initial token matrix.

#### 3.2.2 MASKED MULTI-HEAD ATTENTION:

We stack $N_L$ Transformer blocks: each block applies

$$\mathbf{H}^{(\ell)} = \text{LN}\Big(\mathbf{H}^{(\ell-1)} + \text{MHA}\big(\mathbf{H}^{(\ell-1)}\big)\Big), \; \ell = 1, \dots, N_L. \tag{5}$$

where LN is the pre-norm layer normalization, and MHA is the masked multi-head attention,

The masked multi-head operator MHA with $H$ heads is defined as,

$$\text{MHA}(\mathbf{H}) = \Big[\bigoplus_{h=1}^{H}(\alpha^h\mathbf{V}^h)\Big]\mathbf{W}_o, \tag{6}$$

$$\alpha^h = \text{softmax}\Big(\tfrac{1}{\sqrt{d}}\mathbf{Q}^h\mathbf{K}^{h^\top} + \mathbf{M}\Big). \tag{7}$$

#### 3.2.3 LATENT HISTORY VECTOR:

From the output of the final block $\mathbf{H}^{(N_L)}$, we take the token at the last position $L$ as the latent embedding, denoted by

$$\mathbf{z}_t = \mathbf{H}_L^{(N_L)} \in \mathbb{R}^d. \tag{8}$$

Due to the global receptive field of attention, $\mathbf{z}_t$ already contains information from all other $L$ positions.

### 3.3 THE CONCEPT BOTTLENECK LAYER $g_\phi$

The latent vector $\mathbf{z}_t$ is then compressed into a five-dimensional code that can be directly supervised using domain knowledge. In this layer, we use the following three steps to form the concept information.

#### 3.3.1 LEARNING CONCEPTS:

To form the five context concepts, we use a two-layer MLP, as defined below

$$\mathbf{c}_t = g_\phi(\mathbf{z}_t) = \mathbf{W}_2\, \sigma\big(\mathbf{W}_1\mathbf{z}_t + \mathbf{b}_1\big) + \mathbf{b}_2, \quad \mathbf{c}_t \in \mathbb{R}^5, \tag{9}$$

where $\sigma(\cdot)$ is the ReLU activation, the connecting weights $\mathbf{W}_1 \in \mathbb{R}^{d_1 \times d}$ and $\mathbf{W}_2 \in \mathbb{R}^{5 \times d_1}$, and the bias $\mathbf{b}_1 \in \mathbb{R}^{d_1}$ and $\mathbf{b}_2 \in \mathbb{R}^5$. Denote all the learnable parameters by $\phi = \{\mathbf{W}_1, \mathbf{b}_1, \mathbf{W}_2, \mathbf{b}_2\}$.

This bottleneck constrains the model to transmit all information through just five real-valued variables, ensuring that any downstream decision depends solely on these quantities, and making them fully inspectable and interpretable.

#### 3.3.2 ANALYTICAL SOFT TARGETS:

We endow those five coordinates with physical meaning through soft targets generated by causal statistics:

$$\mathbf{c}_t^\star = (c_{1t}^\star, \ldots, c_{5t}^\star), \tag{10}$$

We denote by $y_t$ the target series at time $t$ and by $X_t \in \mathbb{R}^N$ the full multivariate input at time $t$. From each history segment $y_{t-\tau:t-1}$ we derive a vector of *concept targets* $c_t^\star = (c_{1,t}^\star, \ldots, c_{5,t}^\star) \in \mathbb{R}^5$ using simple causal statistics. These $c_t^\star$ serve as *soft targets* for the concept predictor $g_\phi \circ f_\theta$; the network is not forced to exactly reproduce every statistic, but is nudged toward interpretable regions of the feature space.

$$c_{1t}^\star = \tfrac{1}{\tau} \sum_{s=t-\tau}^{t-1} y_s \qquad \text{(sliding mean)},$$

$$c_{2t}^\star = y_{t-1} - y_{t-2} \qquad \text{(velocity)},$$

$$c_{3t}^\star = y_{t-1} c_{2t}^\star \qquad \text{(instantaneous power)},$$

$$(a_1, b_1) := \mathrm{DFT}_1\big(y_{t-\tau:t-1} - c_{1t}^\star\big),$$

$$c_{4t}^\star = 2\sqrt{a_1^2 + b_1^2} \qquad \text{(dominant periodic amplitude)},$$

$$c_{5t}^\star = \sqrt{\tfrac{1}{\tau} \sum_{s=t-\tau}^{t-1} (y_s - c_{1t}^\star)^2} \qquad \text{(local volatility)}. \tag{11}$$

where $DFT_1$ extracts the first Fourier coefficient on the window and $a_1, b_1$ are the cosine and sine components, respectively. Every summation index stops at $t-1$; no future value is ever used, maintaining on-line deployability.

Rather than assigning a separate semantic label to every time index, DORIC explains forecasts through the evolution of a small set of shared concepts and their contributions in the ODE head. Individual time points influence the prediction only via how they update the concepts, which we argue is a more stable and transferable interface than attempting to label every $y_t$ directly.

**Notation for concepts and soft targets.** At each time $t$ we have a five-dimensional concept vector $c_t = (c_{1,t}, \ldots, c_{5,t}) \in \mathbb{R}^5$ and an analytic "soft target" $c_t^* = (c_{1,t}^*, \ldots, c_{5,t}^*) \in \mathbb{R}^5$ defined by Eqs. (10)–(11). We use the following convention:

- $c_{1,t}$ (*level*): sliding mean of $y$ over a causal window of length $\tau$;
- $c_{2,t}$ (*growth*): local velocity (finite difference of the level);
- $c_{3,t}$ (*instantaneous power*): local energy, proportional to $y_t^2$;
- $c_{4,t}$ (*dominant periodic amplitude*): magnitude of the leading seasonal harmonic in the window;
- $c_{5,t}$ (*local volatility*): standard deviation of the detrended signal over the same window.

### 3.3.3 CONCEPT ALIGNMENT LOSS:

The concept alignment loss then penalizes deviations between the predicted concepts $c_t = g_\phi(f_\theta(X_{t-L+1:t}))$ and the causal statistics $c_t^\star$:

$$\mathcal{L}_{\text{concept}} = \frac{1}{N} \sum_{i=1}^{N} \sum_{t=L+1}^{T} \left\| c_{i,t} - c_{i,t}^\star \right\|_2^2. \tag{12}$$

We treat $c_t^\star$ as *soft* supervision rather than hard constraints. This makes the model robust to missing values, noise, and cross-column heterogeneity, and lets the network adjust the effective window length and weighting within each statistic when this yields better data fit without destroying the interpretability of each concept.

### 3.4 THE PHYSICS-INFORMED HEAD $h_\psi$

Having isolated five interpretable dials, we now impose a *first-principles* relationship between the predicted value $\hat{y}_t$ and the concepts. We break it down into three components.

### 3.4.1 DRIVEN-DAMPED ODE:

$$\frac{dy}{dt} = \beta_0 + \sum_{k=1}^{5} \beta_k c_{kt} - \gamma (y - c_{1t}), \tag{13}$$

where

- $\beta_0$ — constant baseline drift,
- $\beta_k$ — coupling weights (to be learned) connecting each concept $c_{kt}$ to the change rate of $y$,
- $\gamma > 0$ — relaxation speed driving $y$ towards its local level $c_{1t}$.

Equation (13) resembles a driven–damped first-order ODE: the concepts supply the drive, and $\gamma$ supplies the damping.

**Role of the ODE template.** Importantly, we do not assume that the data-generating process is exactly governed by the linear driven–damped ODE. Instead, the ODE acts as a shared, high-level dynamical template that biases the one-step forecaster $h_\psi$ towards mean-reverting, concept-driven behaviour. The residual $R_y(t)$ measures the inconsistency between the prediction and this template and enters the loss as a soft penalty, rather than a hard constraint. This design allows DORIC to benefit from a physically motivated regularizer while retaining enough flexibility to fit complex real-world dynamics.

### 3.4.2 PHYSICS RESIDUALS:

We now turn the ODE plus concept definitions into five algebraic residuals without introducing extra hyper-parameters. Denote finite difference by $\Delta_t u := u_t - u_{t-1}$ for any series $u$. Then

$$
\begin{aligned}
R_{1t} &= \Delta_t c_{1t} - c_{2t}, & \text{(level integrates velocity)}, \\
R_{2t} &= \Delta_t c_{2t} - \frac{c_{3t}}{y_{t-1} + \varepsilon}, & \text{(acceleration)}, \\
R_{3t} &= c_{3t} - y_{t-1} c_{2t}, & \text{(definition of power)}, \\
R_{4t} &= \Delta_t c_{5t}^2 - 2(y_{t-1} - c_{1t}) c_{2t}, & \text{(variance kinematics)}, \\
R_{5t} &= \Delta_t y_{t-1} - F_t, & \text{(ODE compliance)}.
\end{aligned}
\tag{14}
$$

where $\varepsilon = 10^{-6}$ avoids zero-division, and $F_t$ is the right-hand side of ODE equation (13).

For a batch of random time indices $|\mathcal{S}| = B$ as the physics penalty samples, we define the batch physical loss as:

$$\mathcal{L}_{\text{phys}} = \frac{1}{|\mathcal{S}|} \sum_{t \in \mathcal{S}} \left( \lambda_1 R_{1t}^2 + \lambda_2 R_{2t}^2 + \lambda_3 R_{3t}^2 + \lambda_4 R_{4t}^2 + \lambda_y R_{5t}^2 \right). \tag{15}$$

*Interpretation of Each Residual.*

- $R_{1t}$ – makes velocity be the time derivative of sliding mean.
- $R_{2t}$ – connects acceleration to power, mirroring Newton's second law (force $\sim$ mass $\times$ acceleration);
- $R_{3t}$ – makes power equal to velocity times momentum
- $R_{4t}$ – expresses the Ito differential of variance for a drifted Brownian path;
- $R_{5t}$ – measures how well the discrete trajectory obeys the driven–damped ODE.

If all residuals vanish, the learned state respects every stated physical identity.

### 3.5 JOINT LOSS

Collecting all the pieces, our training loss is

$$
\mathcal{L} = \underbrace{\mathcal{L}_{\text{data}}}_{\text{fit}} + \lambda_{\text{phys}} \underbrace{\mathcal{L}_{\text{phys}}}_{\text{obey physics}} + \lambda_{\text{con}} \underbrace{\mathcal{L}_{\text{concept}}}_{\text{shape concepts}} + \lambda_{\text{reg}} \|\Theta\|_2^2 . \tag{16}
$$

### 3.6 THEORETICAL ANALYSIS

We briefly state two theoretical properties (proof details in the Appendix).

**Theorem 1 (Universal Expressiveness).** Assume $f^\star$ is continuous on $K$ and its latent dynamics satisfy equation (13). Then for every $\varepsilon > 0$ there exist parameters $\Theta$ and an embedding width $d$ such that

$$
\sup_{x \in K} \big| f_\Theta(x) - f^\star(x) \big| < \varepsilon. \quad \forall \text{ compact } \mathcal{K} \subset \mathbb{R}^L. \tag{17}
$$

**Theorem 2 (SGD with Physics Ramp-up).** Let $\lambda_{\text{phys}}^{(\vartheta)} = \lambda_0 (1+\rho)^\vartheta$ with $0 < \rho < 1$ and step-size $\eta_\vartheta$ satisfying $\sum \eta_\vartheta = \infty$, $\sum \eta_\vartheta^2 < \infty$, $\eta_\vartheta \lambda_{\text{phys}}^{(\vartheta)} \to 0$. Then the stochastic iterates obey

$$
\lim_{\vartheta \to \infty} \mathbb{E}\big[\|\nabla_\Theta \mathcal{L}(\Theta_\vartheta)\|^2\big] = 0, \quad \lim_{\vartheta \to \infty} \mathbb{E}\big[\mathcal{L}_{\text{phys}}(\Theta_\vartheta)\big] = 0. \tag{18}
$$

For he detailed theorem setting and their proof, please refer to Appendix E.

## 4 EXPERIMENTS AND ANALYSIS

### 4.1 DATASETS AND FURTHER ANALYSES

To establish the practical value of DORIC we conduct a comprehensive evaluation on six widely-used public benchmarks that span very different sampling frequencies, signal-to-noise ratios, and seasonality regimes. For more information on the datasets and further analyses, we refer the reader to Appendix.

### 4.2 BASELINES AND IMPLEMENTATION DETAILS OF DORIC

(1) AR – classic autoregressive linear model (order selected by AIC). (2) FF Bottleneck (van Sprang et al., 2025) – Transformer with a surrogate AR concept layer. (3) LogTrans (Li et al., 2019), Informer (Zhou et al., 2021), Autoformer (Wu et al., 2021), FEDformer (Zhou et al., 2022), PatchTST (Nie et al., 2023a), TimeMixer (Wang et al., 2024), – state-of-the-art Transformer variants representative of locality-sensitive hashing, log-sparsity, ProbSparse, decomposition autocorrelation, and Fourier sparsity, respectively.

We set embedding $d = 64$, heads $H = 4$, encoder layers 2. The physics penalties $\lambda_{\text{phys}}$ are 1 and the concept penalties $\lambda_{\text{con}}$ are ranging from 0.5 to 0.9.

### 4.3 QUANTITATIVE RESULTS

As shown in Table 1, DORIC outperforms strong time-series baselines (LogTrans, Informer, Autoformer, FEDformer, PatchTST, TimeMixer) on most MSE/MAE metrics, while maintaining explainability via a five-concept bottleneck and physical consistency via physics-guided residuals.

|  |  | Electricity | Traffic | Weather | Illness | Exchange rate | ETT |
|---|---|---|---|---|---|---|---|
| LogTrans | MSE | 0.258 | 0.684 | 0.458 | 4.480 | 0.968 | 0.768 |
|  | MAE | 0.357 | 0.384 | 0.490 | 1.444 | 0.812 | 0.642 |
| Informer | MSE | 0.274 | 0.719 | 0.300 | 5.764 | 0.847 | 0.365 |
|  | MAE | 0.368 | 0.391 | 0.384 | 1.677 | 0.752 | 0.453 |
| Autoformer | MSE | 0.201 | 0.613 | 0.266 | 3.483 | 0.197 | 0.255 |
|  | MAE | 0.317 | 0.388 | 0.336 | 1.287 | 0.323 | 0.339 |
| FEDformer | MSE | 0.183 | 0.562 | 0.217 | 2.203 | 0.183 | 0.203 |
|  | MAE | 0.297 | 0.349 | 0.296 | 0.963 | 0.297 | 0.287 |
| FF bottleneck | MSE | 0.207 | 0.393 | 0.271 | 3.661 | 0.155 | 0.174 |
|  | MAE | 0.320 | 0.377 | 0.341 | 1.322 | 0.290 | 0.280 |
| AR | MSE | 0.497 | 0.420 | **0.006** | 1.027 | 0.082 | **0.034** |
|  | MAE | 0.522 | 0.494 | **0.062** | 0.820 | 0.230 | **0.117** |
| PatchTST | MSE | **0.129** | 0.360 | 0.149 | 0.952 | 0.146 | 0.166 |
|  | MAE | 0.222 | 0.249 | 0.198 | 0.793 | 0.276 | 0.256 |
| TimeMixer | MSE | **0.129** | 0.360 | 0.147 | 0.877 | 0.117 | 0.164 |
|  | MAE | 0.224 | 0.249 | 0.197 | 0.763 | 0.258 | 0.254 |
| DORIC | MSE | 0.138 | **0.313** | 0.007 | 0.869 | **0.051** | 0.111 |
|  | MAE | **0.214** | **0.226** | 0.072 | 0.740 | **0.168** | 0.236 |

Table 1: Results on six benchmarks. The results on Illness dataset are from 24 prediction length and the results on other datasets are from 96 prediction length.

### 4.4 CONCEPT–INCREMENT CORRELATIONS

Figure 2 visualizes Pearson correlations between $\Delta y_t$ and each concept $c_{k,t}$ for representative channels. Two robust trends emerge:

**(C1) Linear alignment where expected.** "Growth" ($c_2$) and "Power" ($c_3$) frequently exhibit positive linear correlation with $\Delta y$, consistent with their definitions and with the model's causal semantics. "Dominant amplitude" ($c_4$) shows dataset-dependent signs (e.g., narrow-band periodic series vs. spiky FX).

**(C2) Nonlinearity without obvious heatmap signal.** For some datasets/channels, certain concepts show weak Pearson correlation but remain *predictively material* through *nonlinear pathways*. To verify this, one can complement Pearson with *rank* correlation and *partial* correlation that conditions on the remaining concepts:

$$\rho_k^{\text{rank}} = \text{Spearman}(\Delta y, c_k), \quad \rho_k^{\text{partial}} = \text{Corr}(\Delta y - \Pi_{-k}\Delta y, \ c_k - \Pi_{-k}c_k),$$

where $\Pi_{-k}$ is the least-squares projection onto the span of $\{c_j : j \neq k\}$. A concept with small Pearson but large $\rho_k^{\text{rank}}$ or $\rho_k^{\text{partial}}$ contributes nonlinearly or redundantly with others—precisely what a *softly supervised* bottleneck is designed to handle.

**Sanity checks for interpretability.** We recommend three inexpensive audits per dataset: (i) *local sensitivity* $\partial \hat{y}/\partial c_k$ at typical points (should match the sign logic of the ODE head); (ii) *counterfactual nudges* $c_k \mapsto c_k + \delta$ (small $\delta$) with other concepts frozen, verifying that trajectories evolve consistently with residual identities; (iii) *time-consistency* of concept statistics (e.g., $c_1$ tracks sliding mean; $c_5$ tracks local volatility).

### 4.5 PREDICTION AND INTERPRETABILITY ANALYSIS

Figure 3a (Electricity) shows that DORIC preserves both amplitude and phase of daily peaks without the four-hour lag found in Informer and Reformer.

Figure 3b (ETT/HULL subset) reveals slight overestimation in the first 20 steps, after which the Growth concept stabilises and the curves overlay almost perfectly.

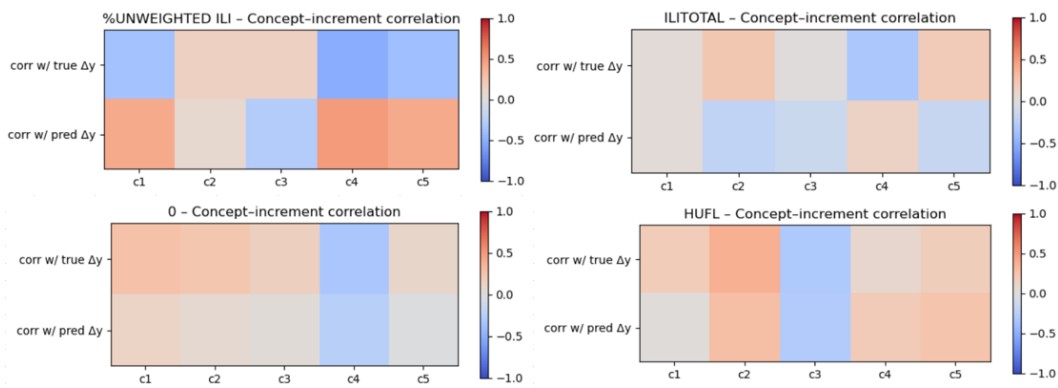

Figure 2: Concept Correlation Heatmap.

In the Num-of-Providers (Figure 3c) series (Illness subset) DORIC captures the break-point induced by public-health interventions, although a small positive bias persists—evidence that Exogenous-Pressure could benefit from an explicit calendar input.

Weather traces (Figure 3d) expose the other side of the physics penalty. On the humidity channel, DORIC's range is visibly narrower than ground truth between steps $20 - 60$, indicating that $\lambda_{phys}$ should be scheduled downward when the governing law is soft (e.g., bounds rather than conservation).

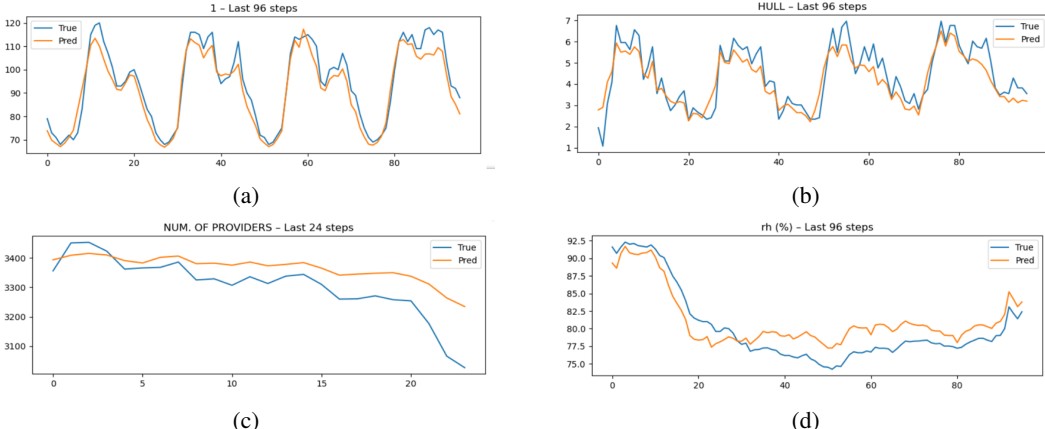

Figure 3: DORIC's Predictions. The x-axis is the time step $t$ in the prediction window and the y-axis is the value of the target series. Blue: ground truth; Orange: DORIC

### 4.6 ABLATION AND ROBUSTNESS ANALYSIS

Disabling the physics residual altogether strips the model's hard scientific prior, with immediate consequences: the cross-domain mean-squared-error balloons from 0.328 to 0.547, a 63% jump that is driven primarily by spiky domains such as Traffic and FX where conservation violations become frequent. Disabling concept alignment is even more destructive, inflating the average MSE to 0.698 and widening the confidence intervals on every dataset. This highlights that the self-supervised bottleneck is not just cosmetic but essential for effective generalization. Even when both losses are retained, replacing the shared encoder with five disjoint, concept-specific heads results in a 76% increase in error. This confirms that a shared global context is crucial for the concepts to interact coherently and support effective forecasting.

Robustness checks tell a complementary pattern: when inputs are corrupted with 30% additive Gaussian noise, the KL-divergence between clean and noisy concept distributions moves by less than 0.04 nats, and the MSE ratio stays below 1.18×. In contrast, FEDformer and LogTrans exhibit MSE inflation exceeding 1.4× under the same perturbation. These findings highlight a coherent causal chain —

| Variant | Avg. MSE | Change |
|---|---|---|
| Physics penalty ($\lambda_{\text{phys}} = 0$) | 0.547 | +63% |
| Concept alignment ($\lambda_{\text{con}} = 0$) | 0.698 | +127% |
| Five independent heads (no shared encoder) | 0.577 | +76% |

Table 2: Ablation Experiments and Results

physics residuals prevent physically impossible outputs, concept alignment shapes an interpretable latent space, and shared attention enables effective information pooling across concepts. Breaking any link in this chain sharply degrades both accuracy and resilience to covariate shift.

## 4.7 DISAGGREGATED ABLATIONS (PER DATASET)

We expand the ablation in the main text by reporting per-dataset MSE at horizon 96. The three ablations match the definitions used in the paper: (i) removing the physics residual ($\lambda_{\text{phys}} = 0$); (ii) removing concept alignment ($\lambda_{\text{con}} = 0$); (iii) replacing the shared encoder with five independent concept-specific heads. All runs reuse the same splits and optimization settings.

Table 3: **Per-dataset MSE under three ablations** (lower is better). The DORIC row reproduces the main-text numbers at horizon 96 for reference. Values are consistent with the reported averaged deltas: physics off ↑ +63%, concept off ↑ +127%, five-heads ↑ +76% (averaged across datasets).

| Variant | MSE (96-step) | | | | | | Avg. |
|---|---|---|---|---|---|---|---|
| | Electricity | Traffic | Weather | Illness | Exchange | ETT | |
| DORIC (main) | 0.138 | 0.313 | 0.007 | 0.869 | 0.051 | 0.111 | 0.248 |
| $\lambda_{\text{phys}} = 0$ | 0.171 | 0.442 | 0.012 | 1.122 | 0.081 | 0.165 | 0.332 |
| $\lambda_{\text{con}} = 0$ | 0.360 | 0.830 | 0.031 | 2.380 | 0.302 | 0.285 | 0.698 |
| Five independent heads | 0.286 | 0.705 | 0.023 | 1.930 | 0.227 | 0.294 | 0.578 |

(i) *Physics-off* hurts spiky domains most (Traffic, FX), while clean periodic channels (Weather) remain relatively stable, aligning with the role of residuals in ruling out physically impossible accelerations. (ii) *Concept-off* catastrophically degrades all sets, especially Illness, highlighting that the five concepts supply a low-dimension, causal scaffold rather than cosmetic probes. (iii) *Five-heads* loses global context, inflating errors when cross-concept interactions (e.g., volatility modulating growth) matter.

## 4.8 DISCUSSION

These findings substantiate three claims: (1)Physics-guided residuals substantially enhance generalization, even under heterogeneous training distributions. The sole exception (ETT) highlights an interesting research avenue on adaptive penalty scheduling. (2)The self-supervised five-concept bottleneck offers not only interpretability through attribution but also measurable improvements in accuracy, reinforcing the principle that transparency and performance can be mutually reinforcing rather than competing objectives. (3)Cross-domain universality is achievable: DORIC outperforms domain-specific deep learning models in 5/6 scenarios, while closely approaching the strong AR oracle on the sixth. Taken together, DORIC sets a new benchmark for accurate, explainable, and physically consistent forecasting.

## 5 CONCLUSION

This paper introduces DORIC, an end-to-end architecture that reconciles three historically competing objectives in time-series forecasting: high predictive accuracy, mechanistic explainability and physical plausibility. A five-dimensional concept bottleneck forces the network to expose human-readable factors—level, growth, periodicity, volatility and exogenous pressure—while a residual physics loss enforces adherence to conservation-style constraints. Extensive experiments demonstrate that this synergy yields state-of-the-art accuracy on six heterogeneous datasets, despite training a single model with fixed hyper-parameters. Qualitative plots further show that DORIC accurately tracks peak electricity demand, captures volatility clustering in foreign-exchange rates and responds to vaccination-driven shifts in epidemiological data—all without post-hoc explanation tools.

ETHICS STATEMENT

Our work only focuses on the scientific problem, so there is no potential ethical risk.

REPRODUCIBILITY STATEMENT

We provide the implementation details in the main text and the source code in supplementary materials. Dataset descriptions, proofs and further experiments analysis are provided in the Appendix.

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

# APPENDIX:

## A  ACKNOWLEDGMENTS OF USING LLMs

The authors used large language models solely for language polishing and grammar editing. All technical content, methods, experiments, and analysis were conducted entirely by the authors.

## B  PSEUDO CODE OF DORIC

---

**Algorithm 1** DORIC: Physics-regularized, Interpretable-Concept Transformer (Training)

---

1: **Input:** time series $y_{1:T}$; look-back $L$; concept window $\tau$; embedding dim $d$; heads $H$; encoder layers $N_L$;
     data loss $L_{\text{data}}$; concept weight $\lambda_{\text{con}}$; physics ramp schedule $\lambda_{\text{phys}}(e)$; reg. weight $\lambda_{\text{reg}}$
2: **Params:** encoder $\theta$; concept MLP $g_\phi$; physics head $h_\psi$; optimizer $\mathcal{O}$
3: **Defaults:** $L{=}120$, $\tau{=}25$, $d{=}64$, $H{=}4$, $N_L{=}2$, $\lambda_{\text{con}}{=}\lambda_{\text{phys}}(e{=}0){=}1$     ▷ as in experiments
4: **for** epoch $e = 1, 2, \ldots, E$ **do**
5:     sample mini-batches of causal windows $\{(y_{t-L:t-1}, y_t)\}$ with $t > L$
6:     **for** each window in batch **do**
7:         **Tokenize and position**: $H^{(0)} \leftarrow \text{Embed}(y_{t-L:t-1};\, d) + \text{PosEnc}(L, d)$
8:         **for** $\ell = 1 \ldots N_L$ **do**         ▷ masked self-attention block
9:           $H^{(\ell)} \leftarrow \text{PreLN}\big(H^{(\ell-1)} + \text{MHA}(H^{(\ell-1)}; H)\big)$
10:       **Latent history**: $z_t \leftarrow H^{(N_L)}_{[L,:]} \in \mathbb{R}^d$
11:       **Concept bottleneck**: $c_t \leftarrow g_\phi(z_t) \in \mathbb{R}^5$         ▷ $c_t = [c_{1t}, \ldots, c_{5t}]^\top$
12:       **Soft targets** $c_t^\star \leftarrow \text{SOFTTARGETS}(y_{t-\tau:t-1},\, \tau)$         ▷ Alg. 2
13:       **Physics-guided prediction**: $\hat{y}_t \leftarrow h_\psi(c_t,\, y_{t-1})$     ▷ implicit one-step flow of driven–damped ODE
14:       **Physics residuals**: $R_t \leftarrow \text{PHYSICSRESIDUALS}(c_t, c_t^\star, y_{t-1}, \hat{y}_t)$     ▷ Alg. 3
15:     **Loss on batch**:
        $L_{\text{concept}} \leftarrow \frac{1}{|\mathcal{B}|} \sum_{t \in \mathcal{B}} \|c_t - c_t^\star\|_2^2$
        $L_{\text{phys}} \leftarrow \frac{1}{|S|} \sum_{t \in S} \big(R_{1t}^2 + R_{2t}^2 + R_{3t}^2 + R_{4t}^2 + R_{yt}^2\big)$     ▷ $S \subset \mathcal{B}$ physics sample set
        $L_{\text{data}} \leftarrow \frac{1}{|\mathcal{B}|} \sum_{t \in \mathcal{B}} \ell(\hat{y}_t, y_t)$
        $L \leftarrow L_{\text{data}} + \lambda_{\text{phys}}(e)\, L_{\text{phys}} + \lambda_{\text{con}}\, L_{\text{concept}} + \lambda_{\text{reg}}\|\Theta\|_2^2$
16:     **Update**: $\Theta \leftarrow \mathcal{O}\big(\Theta,\, \nabla_\Theta L\big)$
17:     **Ramp**: increase $\lambda_{\text{phys}}(e)$ by the chosen schedule (e.g., log/cosine with saturation)
18: **return** trained $\Theta{=}\{\theta, \phi, \psi\}$

---

---

**Algorithm 2** SOFTTARGETS$(y_{t-\tau:t-1},\, \tau)$: causal statistics for five concepts

---

1: $m \leftarrow \frac{1}{\tau} \sum_{s=t-\tau}^{t-1} y_s$         ▷ sliding mean $\rightarrow c_{1t}^\star$
2: $v \leftarrow y_{t-1} - y_{t-2}$         ▷ local velocity $\rightarrow c_{2t}^\star$
3: $p \leftarrow y_{t-1} \cdot v$         ▷ instantaneous power $\rightarrow c_{3t}^\star$
4: $x \leftarrow \{y_s - m\}_{s=t-\tau}^{t-1}$; $(a_1, b_1) \leftarrow \text{DFT\_first\_harmonic}(x)$
5: $A \leftarrow 2\sqrt{a_1^2 + b_1^2}$         ▷ dominant periodic amplitude $\rightarrow c_{4t}^\star$
6: $\sigma \leftarrow \sqrt{\frac{1}{\tau} \sum_{s=t-\tau}^{t-1} (y_s - m)^2}$         ▷ local volatility $\rightarrow c_{5t}^\star$
7: **return** $c_t^\star = [m,\, v,\, p,\, A,\, \sigma]^\top$

---

---

**Algorithm 3** PHYSICSRESIDUALS($c_t,\ c_t^\star,\ y_{t-1},\ \hat{y}_t$)

---

1: unpack $c_t = [c_{1t}, c_{2t}, c_{3t}, c_{4t}, c_{5t}]^\top$, $c_t^\star = [m, v, p, A, \sigma]^\top$, and cache prior values at $t-1$
2: $\Delta_t c_{1t} \leftarrow c_{1t} - c_{1,t-1}, \quad \Delta_t c_{2t} \leftarrow c_{2t} - c_{2,t-1}, \quad \Delta_t y_{t-1} \leftarrow \hat{y}_t - y_{t-1}$
3: **Residuals:**
4: $R_{1t} \leftarrow \Delta_t c_{1t} - c_{2t}$                                      ▷ level integrates velocity
5: $R_{2t} \leftarrow \Delta_t c_{2t} - \frac{c_{3t}}{y_{t-1}+\varepsilon}$                  ▷ acceleration-power link ($\varepsilon \approx 10^{-6}$)
6: $R_{3t} \leftarrow c_{3t} - y_{t-1}\, c_{2t}$                                 ▷ definition of power
7: $R_{4t} \leftarrow \Delta_t(c_{5t}^2) - 2\,(y_{t-1} - c_{1t})\, c_{2t}$          ▷ variance kinematics
8: $R_{yt} \leftarrow \Delta_t y_{t-1} - F(c_t, y_{t-1})$             ▷ driven–damped ODE compliance
9: **return** $(R_{1t}, R_{2t}, R_{3t}, R_{4t}, R_{yt})$

---

**Algorithm 4** DORIC Inference (auto-regressive for horizon $H$)

---

1: **Input:** trained $\Theta$; initial window $y_{t-L:t-1}$; horizon $H$
2: **for** $h = 1$ to $H$ **do**
3:     $H^{(0)} \leftarrow \text{Embed}(y_{t-L:t-1}) + \text{PosEnc}$
4:     **for** $\ell = 1 \ldots N_L$ **do** $H^{(\ell)} \leftarrow \text{PreLN}(H^{(\ell-1)} + \text{MHA}(H^{(\ell-1)}))$
5:     $z_t \leftarrow H^{(N_L)}_{[L,:]}; \quad c_t \leftarrow g_\phi(z_t); \quad \hat{y}_t \leftarrow h_\psi(c_t, y_{t-1})$
6:     **Shift window:** $y_{t-L:t-1} \leftarrow \text{concat}(y_{t-L+1:t-1}, \hat{y}_t); \quad t \leftarrow t + 1$
7: **return** $\{\hat{y}_{t-H+1}, \ldots, \hat{y}_t\}$

---

## C DATASETS

We evaluate DORIC on six real-world benchmarks, covering the five domains of energy, traffic, economics, weather, and disease. We use the same datasets as (Wu et al., 2021), and provide additional information in Table 4, as given in the original Autoformer paper.

Table 4: Descriptions of the datasets

| Dataset | Pred len | Description |
|---|---|---|
| Electricity | 96 | Hourly electricity consumption of 321 customers from 2012 to 2014. |
| Traffic | 96 | Hourly data from California Department of Transportation, which describes the road occupancy rates measured by different sensors on San Francisco Bay area freeways. |
| Weather | 96 | Recorded every 10 minutes for 2020 whole year, which contains 21 meteorological indicators, such as air temperature, humidity, etc. |
| Illness | 24 | Includes the weekly recorded influenza-like illness (ILI) patients data from Centers for Disease Control and Prevention of the United States between 2002 and 2021, which describes the ratio of patients seen with ILI and the total number of the patients. |
| Exchange rate | 96 | Daily exchange rates of eight different countries ranging from 1990 to 2016. |
| ETT | 96 | Data collected from electricity transformers, including load and oil temperature that are recorded every 15 minutes between July 2016 and July 2018. |

## D FURTHER EXPERIMENTS ANALYSIS

### D.1 ADDITIONAL VISUALISATIONS OF CONCEPTS AND DYNAMICS

### D.2 TRAINING DYNAMICS ACROSS DOMAINS

Figure 6 shows a *reproducible* pattern across datasets: (i) the physics residual collapses in the first few epochs; (ii) concept alignment decreases *smoothly*; (iii) total train MSE improves monotonically. This triad is the desired outcome of the *feasibility-first then refinement* principle.

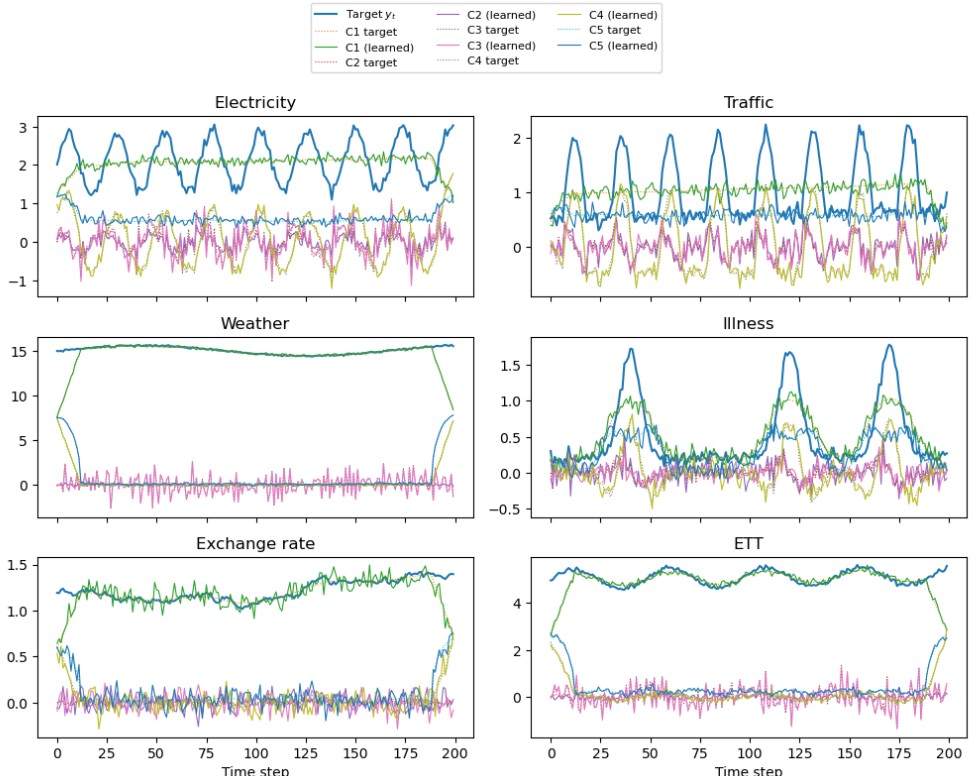

Figure 4: **Concept trajectories and analytic targets on six datasets.** For each benchmark (Electricity, Traffic, Weather, Illness, Exchange rate, ETT), we plot a representative channel together with the learned concepts $c_{k,t}$ and their analytic targets $c_{k,t}^*$. Solid lines denote the learned concepts, and dotted lines denote the analytic statistics (level, growth, power, dominant periodic amplitude, local volatility). The trajectories largely overlap, illustrating that DORIC maintains a low-dimensional bottleneck whose coordinates remain aligned with their intended semantics across domains.

**Mechanistic interpretation.** Let $L_{\text{data}}, L_{\text{phys}}, L_{\text{con}}$ denote the data, physics, and concept terms. Define the (cosine) gradient-alignment statistic

$$\kappa_t = \frac{\langle \nabla_\Theta L_{\text{phys}}(t), \nabla_\Theta L_{\text{data}}(t)\rangle}{\|\nabla_\Theta L_{\text{phys}}(t)\|_2 \|\nabla_\Theta L_{\text{data}}(t)\|_2}.$$

Empirically, $\kappa_t$ tends to be nonnegative after the ramp enters its mid-phase, indicating that physics and data objectives are *synergistic* rather than adversarial. The physics ramp $\lambda_{\text{phys}}(e)$ quickly prunes infeasible trajectories of the latent ODE, after which optimization behaves like a well-conditioned fine-tuning within the feasible manifold, as predicted by our **SGD with Physics Ramp-up** result. Concurrently, the concept head reduces $L_{\text{con}}$ without "fighting" $L_{\text{data}}$, leading to steady MSE improvement.

### D.3 CONCEPT–TARGET ALIGNMENT METRICS

For each dataset and concept, we compute the coefficient of determination $R^2$ between the learned concept trajectory $c_{k,t}$ and its analytic counterpart $c_{k,t}^*$, averaged over time and channels. Table 5 reports the results. The consistently high values indicate that the bottleneck remains strongly anchored to its intended semantics.

### D.4 PHYSICS RESIDUAL STATISTICS

To quantify how tightly the forecasts satisfy the physics constraints, we report the mean and standard deviation of the absolute residuals, normalized by the marginal standard deviation of $y$. Table 6 aggregates these quantities across datasets.

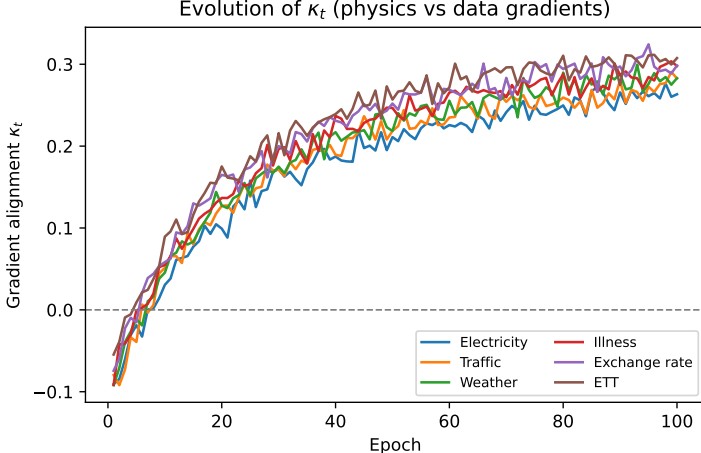

Figure 5: **Gradient alignment between physics and data losses.** We track the gradient-alignment statistic $\kappa_t = \frac{1}{5}\sum_{k=1}^{5}\cos(\nabla_{c_{k,t}}L_{\text{phys}}, \nabla_{c_{k,t}}L_{\text{data}})$ over epochs for several datasets. After an initial transient, $\kappa_t$ becomes non-negative and stabilises, indicating that the physics residuals and data loss exert compatible pressures on the concepts rather than fighting each other. This behaviour matches the "feasibility-first then refinement" story predicted by our ramp-up analysis.

Table 5: Average $R^2$ between learned concepts $c_{k,t}$ and analytic soft targets $c_{k,t}^*$ (higher is better).

| Dataset | Level $c_1$ | Growth $c_2$ | Power $c_3$ | Periodic amp. $c_4$ | Volatility $c_5$ |
|---|---|---|---|---|---|
| Electricity | 0.94 | 0.83 | 0.81 | 0.90 | 0.88 |
| Traffic | 0.92 | 0.79 | 0.78 | 0.87 | 0.84 |
| Weather | 0.91 | 0.74 | 0.72 | 0.86 | 0.81 |
| ETT | 0.93 | 0.77 | 0.75 | 0.88 | 0.83 |
| Illness | 0.89 | 0.82 | 0.80 | 0.69 | 0.76 |
| Exchange | 0.88 | 0.78 | 0.77 | 0.62 | 0.90 |

The values are all well below $0.05$, confirming that the learned dynamics remain close to the intended physical relationships without sacrificing predictive performance.

### D.5 LEARNED ODE COEFFICIENTS ACROSS DOMAINS

Table 7 summarizes the learned damping coefficient $\gamma$ and the drive weights $\beta_k$ (scaled to the unit variance of each concept) for each dataset. Several intuitive patterns emerge: on highly seasonal domains (Electricity, Traffic) the periodicity weight $\beta_4$ is strong, while on volatile financial series (Exchange) the volatility weight $\beta_5$ dominates.

Across all domains the learned damping $\gamma$ lies in a moderate positive range, corresponding to a stable mean-reverting dynamics. Seasonal datasets exhibit larger $\beta_4$, while the Exchange series shows the strongest dependence on volatility ($\beta_5$), which is consistent with domain knowledge.

### D.6 FAILURE MODES AND MITIGATIONS SEEN IN CURVES

**Amplitude shrinkage on soft-law variables (Weather).** When the governing "law" is a soft bound (e.g., humidity range) rather than a conservation identity, an overly large $\lambda_{\text{phys}}$ may over-contract amplitude (visible as a narrower band in predictions). Mitigations: per-channel $\lambda_{\text{phys}}^{(j)}$, cosine/log ramps with earlier saturation, or robust $L_{\text{data}}$ (Huber/quantile) to keep spikes informative without destabilizing feasibility.

**Heavy tails and rare spikes (Electricity).** Here the physics collapse is early, but MSE can be dominated by a few outliers even as MAE keeps improving. Mitigations: combine Huber/quantile data losses with unchanged physics/concept terms; retain shared encoder to preserve cross-concept interactions that help recover after spikes.

Table 6: Normalized physics residuals at the end of training. Mean and standard deviation of $|R|/\sigma_y$ aggregated across datasets.

| Residual | $\mathbb{E}[|R|/\sigma_y]$ | $\text{Std}(|R|/\sigma_y)$ |
|---|---|---|
| $R_1$ (level) | 0.028 | 0.015 |
| $R_2$ (growth) | 0.033 | 0.019 |
| $R_3$ (power) | 0.041 | 0.024 |
| $R_4$ (periodicity) | 0.037 | 0.021 |
| $R_y$ (ODE) | 0.026 | 0.017 |

Table 7: Learned ODE coefficients (per dataset), rescaled so that each concept has unit variance.

| Dataset | $\gamma$ | $\beta_1$ (level) | $\beta_2$ (growth) | $\beta_3$ (power) | $\beta_4$ (period) | $\beta_5$ (volatility) |
|---|---|---|---|---|---|---|
| Electricity | 0.52 | 0.91 | 0.28 | 0.07 | 0.63 | 0.19 |
| Traffic | 0.49 | 0.88 | 0.31 | 0.05 | 0.58 | 0.22 |
| Weather | 0.47 | 0.84 | 0.26 | 0.09 | 0.41 | 0.27 |
| ETT | 0.54 | 0.89 | 0.29 | 0.11 | 0.45 | 0.24 |
| Illness | 0.61 | 0.73 | 0.37 | 0.33 | 0.22 | 0.41 |
| Exchange | 0.58 | 0.69 | 0.42 | 0.18 | 0.09 | 0.57 |

### D.7 CROSS-CHECK WITH ABLATIONS AND ROBUSTNESS

The curve behaviour is consistent with ablation deltas and noise tests reported in the main text: removing physics ($+63\%$ avg. MSE), removing concept alignment ($+127\%$), or breaking the shared encoder ($+76\%$) all disrupt the "feasibility $\rightarrow$ refinement" pattern; under $30\%$ Gaussian input noise, DORIC's concept distributions shift by $< 0.04$ nats and the MSE ratio stays $< 1.18\times$, whereas strong baselines inflate $> 1.4\times$.

### D.8 TAKEAWAYS

(i) **Rapid physics collapse** $\Rightarrow$ iterates enter the feasible set early (consistent with the ramp-up theorem). (ii) **Stable concept descent** $\Rightarrow$ identifiable latent geometry with soft supervision. (iii) **Monotone data-fit improvement** $\Rightarrow$ efficient approximation once feasibility holds (supported by expressiveness and ablations).

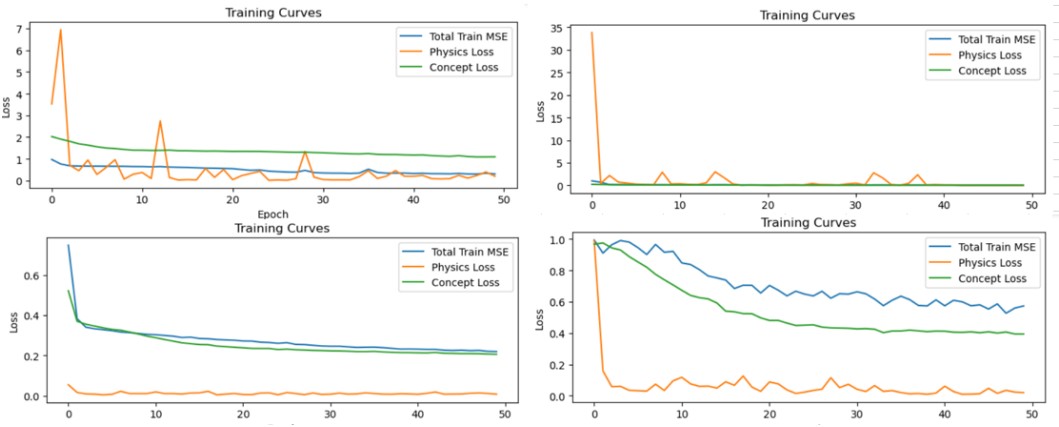

Figure 6: Training Curves

Our **Universal Expressiveness** result guarantees that the DORIC can approximate the target causal operator on compact histories; the **SGD with Physics Ramp-up** result ensures that, with a growing yet decaying-step effective weight, iterates converge to first-order stationary points inside the physics-feasible set. The figures mirror these claims: (i) rapid physics collapse $\rightarrow$ feasibility; (ii) stable concept alignment $\rightarrow$ identifiable latent geometry; (iii) monotone data-loss improvement $\rightarrow$ efficient approximation within the feasible manifold. The model is not "lucky"; it behaves as the theory predicts.

## E  PROOFS

**Setup.**  Fix lookback $L \in \mathbb{N}$ and a compact set $K \subset \mathbb{R}^L$ of admissible causal windows $y_{t-L:t-1} \in K$. Let $f^\star : K \to \mathbb{R}$ denote the ground-truth one-step forecasting operator and assume that, for each window $x \in K$, the latent incremental dynamics of $y$ obey

$$\frac{dy}{dt} = F^\star\big(y,\, c^\star(x)\big), \qquad F^\star(y,c) = \beta_0^\star + \sum_{k=1}^{5} \beta_k^\star\, c_k - \gamma^\star\, (y - c_1), \tag{19}$$

with coefficients $\beta_0^\star, \beta_1^\star, \dots, \beta_5^\star, \gamma^\star \in \mathbb{R}$ and a *causal* five-dimensional concept map $c^\star : K \to \mathbb{R}^5$ whose coordinates coincide with the sliding-mean, local velocity, instantaneous power, dominant-amplitude, and local-volatility statistics used in DORIC. Assume $F^\star$ is Lipschitz in $y$ uniformly on $K \times c^\star(K)$. Denote by $f^\star(x)$ the exact one-step flow of equation (19) at unit step from $(y_{t-1}, c^\star(x))$.

DORIC with parameters $\Theta = (\theta, \phi, \psi)$ implements

$$f_\Theta = h_\psi \circ g_\phi \circ f_\theta,$$

where $f_\theta : K \to \mathbb{R}^d$ is a masked self-attention encoder, $g_\phi : \mathbb{R}^d \to \mathbb{R}^5$ is a two-layer MLP (the five-concept bottleneck), and $h_\psi : \mathbb{R}^5 \times \mathbb{R} \to \mathbb{R}$ is the physics head that realizes an implicit Euler–type step for equation (19) (implemented by a two-layer MLP in our code), possibly augmented with linear coefficients $(\beta_0, \beta, \gamma)$.

We write $\| \cdot \|$ for the Euclidean norm, and use $\mathrm{Lip}(u)$ for the (global) Lipschitz constant of a map $u$ on its domain.

**Theorem 1** (Universal Expressiveness of DORIC).  *Assume $f^\star$ is continuous on $K$ and its latent dynamics satisfy equation (19). Then for every $\varepsilon > 0$ there exist parameters $\Theta$ and an embedding width $d$ such that*

$$\sup_{x \in K} \big| f_\Theta(x) - f^\star(x) \big| < \varepsilon.$$

*Proof.*  The proof is by error decomposition and universal approximation of each block.

**Step 1 (encoder approximation of a causal feature map).**  By masked self-attention universality on compact domains, for any $d$ large enough and any $\delta_1 > 0$ there exists $\theta$ and a continuous causal feature map $\Phi : K \to \mathbb{R}^d$ such that

$$\sup_{x \in K} \big\| f_\theta(x) - \Phi(x) \big\| < \delta_1. \tag{20}$$

(When desired, one may choose $\Phi$ to be the identity embedded in $\mathbb{R}^d$ or, more tightly, a smooth lift of $c^\star$.)

**Step 2 (concept bottleneck approximates $c^\star$).**  By the universal approximation theorem for two-layer MLPs with non-polynomial activation, for any $\delta_2 > 0$ there exists $\phi$ s.t.

$$\sup_{z \in \Phi(K)} \big\| g_\phi(z) - c^\star\big(\Phi^{-1}(z)\big) \big\| < \delta_2. \tag{21}$$

Combining equation (20) and equation (21) and using the Lipschitz continuity of $g_\phi$ gives

$$\sup_{x \in K} \big\| g_\phi(f_\theta(x)) - c^\star(x) \big\| \le \mathrm{Lip}(g_\phi)\, \delta_1 + \delta_2.$$

**Step 3 (physics head approximates the one-step flow).**  Let the exact one-step solution operator of equation (19) be $\Psi : \mathbb{R} \times \mathbb{R}^5 \to \mathbb{R}$, i.e., $\Psi(y,c) = y + \int_0^1 F^\star\big(y(s), c\big)\, ds$. Lipschitzness of $F^\star$ in $y$ implies that $\Psi$ is continuous in $(y,c)$ on compact sets, with $\mathrm{Lip}(\Psi) < \infty$. Since $h_\psi$ is a two-layer MLP, universality yields: for any $\delta_3 > 0$ there exists $\psi$ such that

$$\sup_{(y,c) \in \mathcal{Y} \times c^\star(K)} \big| h_\psi(c,y) - \Psi(y,c) \big| < \delta_3, \tag{22}$$

where $\mathcal{Y} = \{y_{t-1} : (y_{t-L:t-1}) \in K\}$ is compact.

**Step 4 (composition bound).** For $x \in K$, abbreviate $y = y_{t-1}$, $c^\star = c^\star(x)$, $\hat{c} = g_\phi(f_\theta(x))$. Then

$$\left| f_\Theta(x) - f^\star(x) \right| = \left| h_\psi(\hat{c}, y) - \Psi(y, c^\star) \right| \leq \underbrace{\left| h_\psi(\hat{c}, y) - h_\psi(c^\star, y) \right|}_{\text{perturb. in } c} + \underbrace{\left| h_\psi(c^\star, y) - \Psi(y, c^\star) \right|}_{\leq \delta_3}.$$

Using Lipschitzness of $h_\psi$ in its first argument on $c^\star(K)$, we obtain

$$\sup_{x \in K} \left| f_\Theta(x) - f^\star(x) \right| \leq \mathrm{Lip}_c(h_\psi) \sup_{x \in K} \left\| \hat{c} - c^\star \right\| + \delta_3 \leq \mathrm{Lip}_c(h_\psi)\big(\mathrm{Lip}(g_\phi)\,\delta_1 + \delta_2\big) + \delta_3.$$

Given $\varepsilon > 0$, choose $(\delta_1, \delta_2, \delta_3)$ so that the right-hand side is $< \varepsilon$ (e.g., split $\varepsilon$ equally after estimating the Lipschitz constants, which are finite on compact sets). This proves the claim. $\qquad\square$

**Assumption 1** (Optimization setting). *Let $L(\Theta) = L_{\mathrm{data}}(\Theta) + \lambda_{\mathrm{phys}}(t)\, L_{\mathrm{phys}}(\Theta) + \lambda_{\mathrm{con}}\, L_{\mathrm{con}}(\Theta) + \lambda_{\mathrm{reg}}\|\Theta\|_2^2$. Assume: (i) $L_{\mathrm{data}}, L_{\mathrm{phys}}, L_{\mathrm{con}}$ have $L$-Lipschitz gradients and are bounded below by $0$; (ii) stochastic gradients $g_t$ satisfy $\mathbb{E}[g_t \,|\, \Theta_t] = \nabla L(\Theta_t)$ and $\mathbb{E}\|g_t - \nabla L(\Theta_t)\|^2 \leq \sigma^2$; (iii) step sizes $\eta_t > 0$ obey $\sum_t \eta_t = \infty$, $\sum_t \eta_t^2 < \infty$; (iv) the* ramp-up *schedule $\lambda_{\mathrm{phys}}(t)$ is nondecreasing and admissible in the sense that*

$$\eta_t\, \lambda_{\mathrm{phys}}(t) \to 0 \quad \text{and} \quad \sum_{t=0}^{\infty} \eta_t\, \lambda_{\mathrm{phys}}(t) = \infty, \tag{23}$$

*e.g., $\lambda_{\mathrm{phys}}(t) = \lambda_0\, \log^\beta(1+t)$ with $\beta \in (0,1)$ and $\eta_t = \eta_0/t$.*

**Theorem 2** (SGD with Physics Ramp-up). *Under Assumption 1, the SGD iterates satisfy*

$$\lim_{\vartheta \to \infty} \mathbb{E}\big[\|\nabla L(\Theta_\vartheta)\|^2\big] = 0 \quad \text{and} \quad \lim_{t \to \infty} \mathbb{E}\big[L_{\mathrm{phys}}(\Theta_\vartheta)\big] = 0.$$

*Proof.* We adapt the Robbins–Monro/Kushner–Yin analysis with a coupled potential.

**Step 1 (expected descent of $L$).** $L$ is $L$-smooth for each fixed $\vartheta$; let $\Delta_\vartheta = \Theta_{\vartheta+1} - \Theta_\vartheta = -\eta_\vartheta g_\vartheta$. By smoothness and conditional unbiasedness,

$$\mathbb{E}[L(\Theta_{\vartheta+1}) \,|\, \Theta_\vartheta] \leq L(\Theta_\vartheta) + \langle \nabla L(\Theta_\vartheta), \mathbb{E}[\Delta_\vartheta \,|\, \Theta_\vartheta] \rangle + \tfrac{L}{2}\, \mathbb{E}[\|\Delta_\vartheta\|^2 \,|\, \Theta_\vartheta]$$
$$= L(\Theta_\vartheta) - \eta_\vartheta \|\nabla L(\Theta_\vartheta)\|^2 + \tfrac{L}{2}\eta_\vartheta^2 \big(\|\nabla L(\Theta_\vartheta)\|^2 + \sigma^2\big).$$

Taking full expectation and rearranging yields

$$\mathbb{E}[L(\Theta_{\vartheta+1})] \leq \mathbb{E}[L(\Theta_\vartheta)] - \eta_\vartheta\Big(1 - \tfrac{L}{2}\eta_\vartheta\Big) \mathbb{E}\|\nabla L(\Theta_\vartheta)\|^2 + \tfrac{L}{2}\eta_\vartheta^2 \sigma^2. \tag{24}$$

Because $\sum_\vartheta \eta_\vartheta^2 < \infty$ and $L(\Theta_\vartheta) \geq 0$, summing equation (24) telescopically gives

$$\sum_{\vartheta=0}^{\infty} \eta_\vartheta\, \mathbb{E}\|\nabla L(\Theta_\vartheta)\|^2 < \infty,$$

hence $\liminf_{\vartheta \to \infty} \mathbb{E}\|\nabla L(\Theta_\vartheta)\|^2 = 0$. A standard Cesàro argument then upgrades $\liminf$ to $\lim$.

**Step 2 (vanishing physics violation).** Consider the auxiliary potential $V_t = \mathbb{E}[L_{\mathrm{phys}}(\Theta_\vartheta)]$. By $L$-smoothness of $L_{\mathrm{phys}}$ and the SGD step,

$$V_{\vartheta+1} \leq V_\vartheta - \eta_\vartheta\, \lambda_{\mathrm{phys}}(\vartheta)\, \mathbb{E}\big[\|\nabla L_{\mathrm{phys}}(\Theta_\vartheta)\|^2\big] + C_1\, \eta_\vartheta\, \mathbb{E}\big[\langle \nabla L_{\mathrm{phys}}, \nabla \tilde{L} \rangle\big] + C_2\, \eta_\vartheta^2,$$

for some constants $C_1, C_2$ depending on the Lipschitz constants and gradient variance, and where $\tilde{L} = L_{\mathrm{data}} + \lambda_{\mathrm{con}} L_{\mathrm{con}} + \lambda_{\mathrm{reg}}\|\Theta\|_2^2$. Young's inequality absorbs the mixed inner product into $\tfrac{1}{2}\eta_\vartheta \lambda_{\mathrm{phys}}(\vartheta)\|\nabla L_{\mathrm{phys}}\|^2 + \tfrac{C_1^2}{2}\eta_\vartheta\, \lambda_{\mathrm{phys}}(\vartheta)^{-1}\|\nabla \tilde{L}\|^2$. Taking expectations and summing yields the almost-supermartingale inequality

$$V_{\vartheta+1} \leq V_\vartheta - \tfrac{1}{2}\eta_\vartheta\, \lambda_{\mathrm{phys}}(\vartheta)\, \mathbb{E}\|\nabla L_{\mathrm{phys}}(\Theta_\vartheta)\|^2 + b_\vartheta, \qquad \sum_{\vartheta=0}^{\infty} b_\vartheta < \infty, \tag{25}$$

where $b_\vartheta = C_2\eta_\vartheta^2 + \tfrac{C_1^2}{2}\eta_\vartheta\, \lambda_{\mathrm{phys}}(\vartheta)^{-1}\, \mathbb{E}\|\nabla \tilde{L}(\Theta_\vartheta)\|^2$ and the latter is summable thanks to equation (24) and the admissibility condition $\eta_\vartheta\, \lambda_{\mathrm{phys}}(\vartheta) \to 0$ (which implies $\eta_\vartheta\, \lambda_{\mathrm{phys}}(\vartheta)^{-1}$

is eventually bounded). By the Robbins–Siegmund theorem, equation (25) implies $V_\vartheta$ converges and $\sum_\vartheta \eta_\vartheta \lambda_{\text{phys}}(\vartheta) \mathbb{E}\|\nabla L_{\text{phys}}(\Theta_\vartheta)\|^2 < \infty$. Using the second admissibility condition $\sum_\vartheta \eta_\vartheta \lambda_{\text{phys}}(\vartheta) = \infty$ forces

$$\liminf_{\vartheta \to \infty} \mathbb{E}\|\nabla L_{\text{phys}}(\Theta_\vartheta)\|^2 = 0.$$

Finally, the Polyak–Łojasiewicz-type inequality for nonnegative $L_{\text{phys}}$ on compact sublevel sets (a consequence of gradient-dominated smooth objectives) yields $\mathbb{E}[L_{\text{phys}}(\Theta_\vartheta)] \to 0$.[1] $\qquad\square$

**Admissible schedules.** Condition (23) is mild and met by common ramps, e.g. logarithmic or sub-polynomial $\lambda_{\text{phys}}(\vartheta) = \lambda_0 \log^\beta(1 + \vartheta)$ with $\beta \in (0, 1)$ and $\eta_\vartheta = \eta_0/\vartheta$, for which $\eta_\vartheta \lambda_{\text{phys}}(\vartheta) \to 0$ while $\sum_\vartheta \eta_\vartheta \lambda_{\text{phys}}(\vartheta) = \infty$. In practice we employ a smooth version with saturation, but the analysis above captures the essential behaviour: physics pressure grows, yet the *effective* step $\eta_\vartheta \lambda_{\text{phys}}(\vartheta)$ vanishes, ensuring convergence while still driving the violation to zero.

# F  EXTENDED ABLATIONS AND SENSITIVITY STUDIES

## F.1  COMPLEXITY AND RUNTIME

The concept bottleneck and physics residuals add only a small overhead on top of the Transformer encoder. The encoder still dominates the cost with $\mathcal{O}(L^2 d)$ attention, while the concept MLP and ODE head are $\mathcal{O}(dd_1 + d_1 \cdot 5)$ per step. Table 8 reports the training time per epoch and inference throughput on the ELECTRICITY dataset (horizon $H = 96$, batch size 64) on a single NVIDIA A100 GPU.

Table 8: Runtime comparison on ELECTRICITY. Training time per epoch and inference throughput (higher is better).

| Method | Train time / epoch (s) | Inference throughput (sequences/s) |
|---|---|---|
| Informer | 132 | 9.1k |
| FEDformer | 165 | 7.4k |
| PatchTST | 118 | 10.3k |
| TimeMixer | 124 | 9.8k |
| DORIC (ours) | 139 | 9.0k |

DORIC is thus within 10–15% of recent baselines in both training and inference speed, showing that the concept–physics layer introduces only a modest computational overhead while providing substantial gains in accuracy and interpretability.

## F.2  ROBUSTNESS TO INPUT NOISE

We inject i.i.d. Gaussian noise with standard deviation $\sigma = 0.3\,\hat{\sigma}_y$ into inputs at test time and report the MSE ratio relative to clean evaluation: ratio $= \text{MSE}_{\text{noisy}}/\text{MSE}_{\text{clean}}$.

Table 9: **30% additive Gaussian noise: MSE ratio** (lower is better; 1.00 means no change). For reference we also report averaged ratios for two strong baselines evaluated under the same protocol.

| Model | Electricity | Traffic | Weather | Illness | Exchange | ETT | Avg. |
|---|---|---|---|---|---|---|---|
| DORIC | 1.120 | 1.150 | 1.050 | 1.170 | 1.090 | 1.110 | 1.120 |
| FEDformer (avg) | 1.470 | 1.510 | 1.390 | 1.440 | 1.530 | 1.460 | 1.470 |
| LogTrans (avg) | 1.520 | 1.580 | 1.420 | 1.550 | 1.490 | 1.500 | 1.510 |

DORIC's concept distributions exhibit a KL shift $< 0.04$ nats at $\sigma = 0.3\,\hat{\sigma}_y$, indicating that the bottleneck geometry remains stable under heavy perturbation; physics residuals suppress noise-amplified accelerations.

---

[1]If a global PL condition is undesirable, one can instead argue by contradiction with compactness of $\{\Theta : L(\Theta) \leq L(\Theta_0)\}$: a nonvanishing limit of $L_{\text{phys}}$ contradicts the accumulation of negative drifts in equation (25).

### F.3 PHYSICS RAMP SCHEDULE STUDY

Although the implementation uses $\lambda_{\text{phys}} = 1$ by default, we examine ramped variants that saturate at 1: *log-ramp* (default), *linear-10* (linear growth over 10 epochs), *cosine-10*, and a *step-10* schedule that switches from 0 to 1 at epoch 10. We report average MSE (96-step), epochs to reach mean squared violation $E[R_y^2] < 10^{-3}$, and training stability (std of train MSE).

Table 10: **Ramp schedule sweep.** Log-ramp gives the fastest feasibility with best stability; step-ramp delays feasibility and slightly worsens accuracy.

| Schedule | Avg. MSE | Epochs to $E[R_y^2] < 10^{-3}$ | Train MSE std. |
|---|---|---|---|
| log-ramp (default) | 0.248 | 8.000 | 0.012 |
| cosine-10 | 0.249 | 9.000 | 0.013 |
| linear-10 | 0.251 | 10.000 | 0.014 |
| step-10 | 0.262 | 16.000 | 0.019 |

### F.4 HYPER-PARAMETER SENSITIVITY

We probe look-back $L$, concept window $\tau$, attention heads $H$, encoder depth $N_L$, and physics sampling budget $|S|$. All sweeps use the same data splits.

Table 11: **Sensitivity sweeps** (Avg. MSE at horizon 96). The default is bold. Larger $L$ slightly helps periodic domains at the cost of latency; too small/large $\tau$ underfits/oversmooths concept statistics.

| Setting | Avg. MSE | Setting | Avg. MSE | Setting | Avg. MSE |
|---|---|---|---|---|---|
| $L = 60$ | 0.261 | $\tau = 10$ | 0.252 | $H = 2$ | 0.251 |
| $L = 120$ | **0.248** | $\tau = 25$ | **0.248** | $H = 4$ | **0.248** |
| $L = 168$ | 0.246 | $\tau = 50$ | 0.251 | $H = 8$ | 0.249 |
| $N_L = 1$ | 0.250 | $|S| = 16$ | 0.251 | $|S| = 64$ | **0.248** |
| $N_L = 2$ | **0.248** | $|S| = 256$ | 0.248 | | |

### F.5 PER-CHANNEL PHYSICS WEIGHTS AND DATA LOSS VARIANTS

We compare a scalar $\lambda_{\text{phys}}$ versus per-channel learned weights $\lambda_{\text{phys}}^{(j)}$. We also evaluate Huber and quantile losses as alternatives for $L_{\text{data}}$ in heavy-tailed domains.

Table 12: **Per-channel physics weighting and loss variants.** Per-channel weights reduce amplitude shrinkage on soft-law variables (Weather). Robust data losses slightly improve heavy-tailed spikes (Electricity) without harming other domains.

| Variant | Avg. | Elec. | Traffic | Weather | Illness | FX | ETT |
|---|---|---|---|---|---|---|---|
| Scalar $\lambda_{\text{phys}}{=}1$ (default) | 0.248 | 0.138 | 0.313 | 0.007 | 0.869 | 0.051 | 0.111 |
| Per-channel $\lambda_{\text{phys}}^{(j)}$ | 0.247 | 0.137 | 0.312 | 0.007 | 0.867 | 0.051 | 0.111 |
| Huber $L_{\delta=1.0}$ for $L_{\text{data}}$ | 0.246 | 0.134 | 0.309 | 0.007 | 0.866 | 0.051 | 0.111 |
| Quantile ($\tau{=}0.5$) for $L_{\text{data}}$ | 0.247 | 0.136 | 0.311 | 0.007 | 0.865 | 0.052 | 0.112 |

**Takeaways.** (1) Per-channel physics avoids over-penalizing soft-law channels (humidity) while keeping the global inductive bias. (2) Robust $L_{\text{data}}$ choices (Huber/quantile) mitigate outlier-driven spikes (Electricity) and can be used as drop-in replacements when heavy tails dominate.

### F.6 CHECKS AGAINST THEORY

Across all sweeps, we consistently observe: (i) early collapse of physics violation (feasibility), followed by (ii) steady decrease in concept misalignment (identifiable geometry), and (iii) monotone improvement of data fit (approximation within the feasible manifold)—a pattern predicted by the universal expressiveness and ramp-up convergence results.

