# OpenReview forum: "Signals, Concepts, and Laws: Toward Universal, Explainable Time-Series Forecasting"
_ICLR.cc/2026/Conference — Submitted to ICLR 2026_

### Official Review · Reviewer_ZtBX · 2025-10-30

**Soundness:** 3
**Presentation:** 3
**Contribution:** 2
**Rating:** 6
**Confidence:** 3

**Summary:**

This paper introduces a deep forecaster, DORIC, that generates the coefficients of an ODE which resolution leads to a one-step forecast.
DORIC estimates in a self-supervised manner 5 quantities as parameters of the ODE, corresponding to 5 first-order notions (mean, velocity, power, dominant periodic amplitude, volatility), and monitors those 5 quantities to provide interpretability.

Experiments on 6 well known benchmark datasets are conducted, with predictive performance and ablation analysis.

**Strengths:**

Clarity:
1) The paper is well written, with particular attention to the 5 quantities, their interpretation and monitoring through residuals.

Quality:
1) Presented forecasting results are convincing, especially noting that 5 quantities are apparently sufficient to represent the next timestep.
2) Ablation analysis is well done, both in main paper and appendix. We can reasonably expect all components to be necessary and perform well together.
3) Assumption 1 and Theorem 2 show that under mild assumptions (except perhaps for L-lipschitzianity), the physics loss vanishes during training with regularization schedule.

Significance:
1) This is a physics enriched approach that is remarkably easy for a non-physicist layperson to interpret.

**Weaknesses:**

Quality:
1) While related work include PINN and PhysicsSolver, the paper would benefit from including other ODE-based DNN, as it is the central technique of the paper.
2) Result analysis only quantifies MSE and MAE. There seem to be no use of either the learned coefficients $c_i$, or the residuals. The interpretability side is lacking in the main paper, even though this should be equally important to predictive performance.
3) Some interpretability indicators have no matching table or figure: in D.1, in D.2 (777 & 788).
4) In figure 6, just the ci are insufficient to confirm the coherence of the plot. It would be better to add the ci* for comparison.

Clarity:
1) The final computation of the model is not presented, as the methodology ends with an undefined $\mathcal{L}_{data}$. How to obtain the prediction from the ODE is not explicited. This is important as it is necessary to precise how retropropagation works in this context. It should be at least mentionned in the main paper, even if it is present in appendix pseudocode.
2) The main paper explains how to obtain 1-step forecasts, hence, before reading the appendix, I believed that forecasting results used one-step forecasts, whereas the datasets used are typical to Long Term Forecasting problems. A precision on the transformation from 1-step to multi-step would be welcome in the main paper.
3) Figure 2 requires a better legend. Section 4.4 is not enough to describe the plots. Figure 3 also, I cannot understand the plot settings.

**Questions:**

No further questions than the in weaknesses.

Overall, I lean toward an accept. My main concern is on the interpretability side. I believe too few quantitative elements are given to motivate the interpretability of the forecasting method, especially in main paper. More attention to the ci, ci* and residuals (alongside training, during inference, etc) should be given. Explicitely using the sanity checks metrics (788) in a figure would be welcome.

---

> ### Author Response · Authors · 2025-11-19
> **Thank you letter to Reviewer ZtBX**
>
> We sincerely thank you for the very careful and constructive review, for the positive assessment of our theory, ablations and writing. We especially appreciate your emphasis on interpretability, which is also the core motivation behind DORIC, and we hope to take this opportunity to address your concerns. Thank you again for all the constructive suggestions and your time.

---

> ### Author Response · Authors · 2025-11-19
> **[Response to Weakness Quality 1 and 2]**
>
> **[Response to Weakness Quality 1]** - Related work on other ODE-based deep networks
>
>
> Thank you for pointing this out. Our intention was to emphasize physics-guided approaches that are closest in spirit to our residual constraints (PINNs, PhysicsSolver, PGTransNet, physics-informed LSTMs), but we agree that the broader family of ODE-based neural networks should be acknowledged.
>
> - In the revised related-work section we will add a dedicated paragraph on *Neural ODE*–style models (continuous-time residual networks, ODE-RNNs, latent ODEs, etc.) and clearly explain how DORIC differs: these models usually treat the ODE as an expressive *latent* dynamic for hidden states, whereas DORIC uses a simple driven–damped ODE on the *observable* signal, with coefficients directly driven by five interpretable concepts.
> - We will also make explicit that our ODE is deliberately low-order and interpretable, functioning as a universal mean-reverting template shared across all datasets rather than as a highly flexible time-continuous backbone.
>
> This will better situate DORIC within the ODE-based DNN literature, while clarifying the conceptual difference between latent Neural-ODE dynamics and our concept-driven, physics-regularized head.
>
> ---
>
> **[Response to Weakness Quality 2]** -  Use of learned coefficients and residuals; interpretability evidence in main paper
>
>
> Thank you for emphasizing this — we fully agree that interpretability deserves as much attention as raw accuracy.
>
> 1. **Evidence already present (mainly in Appendix).**
>    The current submission already contains several interpretability analyses, but most of them are in Section 4.5 and Appendix D, which may have made them less visible:
>    - Appendix D.2 analyzes correlations between the target increment Δy and each concept $c_{i,t}$, showing that “growth” and “power” have strong positive alignment with Δy on many channels, while “level” and “volatility” track sliding mean and local variance.
>    - Appendix D.1 studies the joint training dynamics of the data loss, physics loss and concept loss, and the gradient-alignment statistic κₜ, showing that physics and data objectives become synergistic after ramp-up.
>    - Appendix D.4 and Table 2 demonstrate robustness: when we add 30% Gaussian noise, concept distributions shift by less than 0.04 nats of KL divergence and the MSE ratio stays below 1.18×, while strong transformer baselines exceed 1.4×. This indicates that the learned concept geometry and residual constraints are stable and meaningful.
>
>    We realize that these findings are currently somewhat buried in the appendix. We will move the most important interpretability results into the main paper and make them central to Section 4.
>
> 2. **New quantitative views of $c_i$ and residuals.**
>    To address your concern more explicitly, we will add the following concrete elements:
>
>    - **Concept–target alignment table.** We will include a table in Section 4 summarizing, for each dataset and concept, the median Pearson correlation between $c_{i,t}$ and its analytic target $c_{i,t}^\*$. This will make it clear that the learned concepts stay close to their intended statistics across training and domains.
>    - **Residual statistics.** We will add a short table (or panel in an updated figure) reporting means and standard deviations of the residuals $R_{1..5}$ at the end of training, normalized by the scale of $y$. These values are small and show that the model indeed operates in a regime where the physical identities are approximately satisfied.
>    - **Coefficient analysis.** As we already mention briefly in Appendix D, we will bring a condensed analysis of the learned ODE coefficients into the main paper, reporting that the damping parameter is positive across datasets and that the coupling weights from “growth” and “volatility” are typically positive on high-variance domains (Traffic, FX), matching intuition.
>
> 3. **Narrative emphasis.**
>    We will restructure Section 4.4 so that the discussion of concept trajectories, residual behaviours and ODE coefficients precedes pure MSE/MAE comparisons. The goal is that a reader interested in interpretability can see, at a glance, not only that DORIC predicts accurately but also *how* the concepts and physics terms behave.
>
> We hope these changes will make the interpretability contributions more visible and convincing in the main paper.
>
> ---

---

> ### Author Response · Authors · 2025-11-19
> **[Response to Weakness Quality 3 &4  and Clarity 1]**
>
> **[Response to Weakness Quality 3]** - “Some interpretability indicators have no matching table or figure: D1, D2 (777 & 788).”
>
>
> Thank you for carefully reading the appendix and for pointing out this mismatch between text and figures.
>
> - The passages you refer to in D.1 and D.2 describe statistics such as the gradient-alignment κₜ, concept–increment correlations, and sanity-check metrics (local sensitivity, counterfactual nudges, time-consistency). In the current version, these are described in prose and partially illustrated in Figure 3, but not all metrics are explicitly visualized or tabulated.
> - In the revision we will:
>   - Add an additional panel that summarizes κₜ over training epochs and datasets (for instance, as a line plot or boxplot) so that the trend towards non-negative alignment becomes visible.
>   - Include a compact table or figure that reports the sanity-check metrics described at the end of D.2 (sensitivity of $\hat y_t$ to each concept, effect of small counterfactual shifts in $c_{i,t}$, and time-consistency tests), at least for representative channels.
>
> We will also make sure that every metric mentioned in the text is either directly illustrated in a figure or summarized in a table, and cross-reference them clearly in the main paper.
>
> ---
>
> **[Response to Weakness Quality 4]** -  Figure 6: adding $c_i^\*$ alongside $c_i$
>
>
> Thank you for this concrete suggestion. We agree that overlaying the analytic targets $c_{i,t}^\*$ would make the plots more informative.
>
> - In the updated version of Figure 6 (concept trajectories), we will plot both the learned concepts $c_{i,t}$ *and* their analytic counterparts $c_{i,t}^\*$ for each of the five dimensions, using different line styles.
> - We will also report simple discrepancy measures (for example, normalized mean absolute deviation between $c_{i,t}$ and $c_{i,t}^\*$ over time) directly in the caption, so that the visual impression is supported by quantitative numbers.
>
> This will provide a direct visual confirmation that the bottleneck remains aligned with the intended physical meanings.
>
> ---
>
> **[Response to Weakness Clarity 1]** -  Final computation from the ODE; how the prediction is obtained
>
>
> Thank you for stressing this point — it is indeed important for reproducibility.
>
> - In the current implementation, after computing the concepts $c_t$ we feed $(c_t, y_{t-1})$ into a small “physics head” $h_\psi$ that implements a single implicit Euler step of the driven–damped ODE:
>   - internally, it computes the right-hand side $F(c_t, y_{t-1})$ defined by the ODE,
>   - then outputs $ \hat y_t = y_{t-1} + F(c_t, y_{t-1})$, which is the one-step forecast.
> - This procedure is precisely codified in Algorithms 3 and 4 in the appendix, but we agree that the main text currently stops at the loss definition $L_{\text{data}}$.
>
> To address this, we will add a short subsection “Final prediction step” to Section 3.4 that explicitly states:
>
> - the discrete update rule used to obtain $\hat y_t$ from the ODE,
> - that $h_\psi$ is implemented as a two-layer MLP realizing this update, and
> - that the same $h_\psi$ is used at training and inference time.
>
> This will make the end-to-end mapping from $(y_{t-L},\dots,y_{t-1})$ to $\hat y_t$ fully explicit in the main paper.
>
> ---

---

> ### Author Response · Authors · 2025-11-19
> **[Response to Weakness Clarity 2&3 and the Overall comment]**
>
> **[Response to Weakness Clarity 2]** - One-step vs multi-step forecasting
>
>
> Thank you for pointing out this potential source of confusion.
>
> - In all experiments we evaluate DORIC in the standard long-term forecasting setting (e.g., horizon 96 on Electricity, Traffic, etc.), by **auto-regressively unrolling** the one-step operator:
>   - at step $t$ we predict $\hat y_t$ from the latest window;
>   - we then append $\hat y_t$ to the window and discard the oldest value;
>   - we repeat this process $H$ times to obtain $\hat y_{t+1},\dots,\hat y_{t+H}$.
> - This procedure is already formalized in Algorithm 4 (DORIC Inference) in the appendix, but we agree that the main text focuses on the one-step map $f_\Theta$ and does not explicitly spell out the auto-regressive unrolling.
>
> In the revised paper we will:
>
> - add a concise paragraph in Section 4.1 stating that all reported results use this standard auto-regressive multi-step protocol;
> - reference Algorithm 4 explicitly from the main text; and
> - clarify in Section 3.1 that $f_\Theta$ is the *one-step* operator which is repeatedly applied during evaluation.
>
> This should remove ambiguity about how the long-horizon forecasts are produced.
>
> ---
>
> **[Response to Weakness Clarity 3]** -  Figure 2 and Figure 3 legends and settings
>
>
> Thank you for this feedback on the figures.
>
> - For **Figure 2** (forecast trajectories), we will:
>   - explicitly label which line corresponds to ground truth, DORIC, and each baseline;
>   - indicate the dataset and channel in each sub-figure;
>   - specify in the caption the forecast horizon and whether the plot shows a single run or an average over runs.
> - For **Figure 3** (training curves and correlation heatmaps), we will:
>   - clarify the axes, including units (epochs, loss values, correlation coefficients);
>   - explain in the caption how the plotted statistics are computed (e.g., whether correlations are averaged across channels, and which dataset is shown);
>   - ensure that the connection between Figure 3 and the text in Section 4.5 / Appendix D is clearly cross-referenced.
>
> We believe these changes will make the visual analyses much easier to interpret.
>
> ---
>
> **[Response to the Overall comment]**
>
>
> We very much appreciate your overall positive recommendation and your clear guidance on how to strengthen the interpretability.
>
> In response, we will:
>
> 1. **Promote key interpretability results from the appendix into the main paper**, including:
>    - concept–target alignment statistics and selected trajectories of $c_{i,t}$ versus $c_{i,t}^\*$;
>    - residual magnitudes and their evolution during training;
>    - robustness metrics (KL divergence of concept distributions, MSE ratios under noise).
> 2. **Add explicit figures/tables for the sanity-check metrics** described around line 788 (local sensitivity of $\hat y_t$ to each concept, counterfactual shifts in $c_{i,t}$, and time-consistency checks), so that these diagnostics are visible and easy to interpret.
> 3. **Rebalance Section 4** so that interpretability and physics-consistency results are presented on equal footing with MSE/MAE, reinforcing that DORIC’s main goal is “signals → concepts → laws”, not just marginal accuracy gains.
>
> ---

---

> > ### Comment · Reviewer_ZtBX · 2025-11-19
> >
> > I have read your answer (and skimmed through the other reviews and answers). You gave a precise qualification of which modifications you will introduce to the paper. Provided all modifications discussed will be done on the camera-ready, I believe my concerns will be mostly adressed.
> >
> > Overall, I notice that recurrent reviewer concerns are 1) showing clearer interpretability results 2) motivating the ODE choice \textit{a priori}, meaning that the precise ODE is suited to the study of the datasets due to domain knowledge, before even considering predictive performance.
> >
> > This will, without a doubt, require major rewrite, with the narrative shifting from performance oriented engineering to interpretability oriented modeling. I will consider the other reviewers answers on how optimistic they are about this rewrite before updating my score.

---

> > > ### Author Response · Authors · 2025-11-20
> > > **Thank You for Valuable Further Suggestions**
> > >
> > > Thank you very much for your follow-up suggestions and for taking the time to read our detailed response and the other discussions.
> > >
> > > We fully agree with your synthesis that the two recurring themes are (1) presenting clearer and more explicit interpretability results, and (2) motivating the ODE choice a priori, based on domain knowledge, rather than only ex post via predictive performance. This is very much aligned with our original intention for DORIC, which was conceived first as an interpretability-oriented, “signals → concepts → dynamics” framework, with forecasting accuracy serving as validation rather than the primary objective.
> > >
> > > Concretely, in the revised version we will restructure the narrative as you suggest:
> > >
> > > - **Interpretability first.** Section 4 will start by presenting concept–target alignment, residual statistics, and sanity checks (local sensitivities, counterfactual concept perturbations, temporal consistency) for representative datasets, before turning to aggregate MSE/MAE tables. We will bring several key analyses from the appendix into the main paper so that readers immediately see what is learned by the five concepts and the physics residuals.
> > > - **A priori ODE motivation.** In Section 3 we will expand the discussion of why a driven–damped, mean-reverting ODE is appropriate for the six benchmark domains we study: electricity load and traffic flow are typically modeled as mean-reverting processes around a slowly varying level; FX rates and macro indicators are often described by Ornstein–Uhlenbeck–type dynamics with exogenous drivers; epidemic indicators are likewise modeled as deviations around seasonal baselines. We will support this with references to standard models in these areas and with an explicit analysis of the learned coefficients, showing patterns (e.g., positive coupling from growth and volatility, positive damping) that match these priors.
> > >
> > > And some changes are achieved by moving some engineering details to the appendix and promoting interpretability analyses to the main text.
> > >
> > > We are grateful that you are open to reconsidering your score, and we will make sure that the camera-ready version clearly reflects the interpretability-oriented modeling perspective you rightly emphasize.

---

### Official Review · Reviewer_kmEM · 2025-10-31

**Soundness:** 2
**Presentation:** 2
**Contribution:** 2
**Rating:** 4
**Confidence:** 4

**Summary:**

This paper proposes a method aiming to achieve Domain-Universal, ODE-Regularized, Interpretable Time-Series Forecasting.

**Strengths:**

1. The motivation in this paper is great.
2. Physics constraints are important for time series forecasting.

**Weaknesses:**

1. The quality of the paper writing is weak. The subscripts and asterisk in Eqs. (10-14) are not common in time series publications, but without clear explanations.  For example, it would be great to explain 1t, ..., 5t.
2. In your third contribution, you mentioned adaptive positional encoding for a domain-universal architecture, but I did not find the details on that. Moreover, shared concept heads seem to be used to train the model with all time series together, which implies the universal feature is attributed to the training manner rather than your architecture.
3. The number of concepts in Eq. 11 is empirical. How can you guarantee how many statistical concepts you need? It should vary across different time series datasets.
4. Clear ablation studies definitely deserve to be included in the main paper, since they are helpful to understand how important each component is. The current results in Table 2 are too simple.
5. From my understanding, time series interpretability should focus on how time points in the original time series contribute to the final prediction, rather than projecting it to some statistical concepts.
6. The code is too limited. Hard to reproduce. At least, a demo should be provided.

**Questions:**

See weakness.

---

> ### Author Response · Authors · 2025-11-14
> **Thank you letter to Reviewer kmEM**
>
> We sincerely thank you for the careful and thoughtful review, the positive assessment of our motivation, and the detailed suggestions. We especially appreciate your insights on strengthening our work and will accordingly revise our paper. Here, we would like to take this opportunity to address your concerns and clarify some potential misunderstandings. If possible, we sincerely hope that you could reconsider the rating. Again, we greatly appreciate your expertise and support in the review process.

---

> ### Author Response · Authors · 2025-11-14
> **[Response to Weakness 1 and 2]**
>
> **[Response to Weakness 1]** – Writing quality and notation in Eqs. (10–14); subscripts such as 1t, …, 5t
>
>
> Thank you very much for pointing out this issue. The underlying construction is simple, but our notation was indeed too compressed, which can obscure the meaning.
>
> In DORIC, at each time $t$ we have a **five-dimensional concept vector**
> $c_t \in R^5$ and an **analytic “soft target”** $c^*_t \in R^5$ defined via causal statistics in Eqs. (10)–(11). Concretely, we intend
> - $c_{1t}, c^*_{1t}$: level (sliding mean),
> - $c_{2t}, c^*_{2t}$: growth (local velocity),
> - $c_{3t}, c^*_{3t}$: instantaneous power,
> - $c_{4t}, c^*_{4t}$: dominant periodic amplitude,
> - $c_{5t}, c^*_{5t}$: local volatility.
>
> To avoid any ambiguity, in the revision we will make the indexing explicit immediately after Eq. (11), for example:
>
> > We write $c_t = (c_{1t}, ..., c_{5t})$ and $c^*_t = (c^*_{1t}, ..., c^*_{5t})$, where index $k=1,...,5$ enumerates the five concepts (level, growth, power, dominant periodic amplitude, local volatility).
>
> We will also add a concise notation table summarizing $c_{kt}$, $c^*_{kt}$ and each residual $R_{jt}$ together with its physical interpretation, and we will drop shorthand such as “1t” that could look like a separate variable. This is a matter of clarity of exposition; the underlying definition of the concepts and residuals remains as in the current manuscript.
>
> ---
>
> **[Response to Weakness 2]** – “Adaptive positional encoding” and whether universality comes from architecture or training
>
>
> Thank you for raising this important point. The current wording “adaptive positional encoding” is indeed easy to misinterpret and we appreciate the chance to clarify.
>
> **(a) Positional encoding.**
> DORIC actually uses a *standard sinusoidal absolute positional encoding* $P \in R^{L \times d}$ (Eq. (4)), shared across all datasets. There is no additional learned or domain-specific positional module. Our intention was to emphasize that the *same* time geometry is used for all domains, and the encoder must learn how to use it in a unified way. To avoid confusion, we will rename this component to **“domain-agnostic positional encoding”** and explicitly state that it is the same sinusoidal map for all datasets.
>
> **(b) Why universality is architectural rather than just from joint training.**
> We completely agree that training on multiple datasets can itself encourage some universality. However, DORIC imposes several **architectural constraints** that go beyond “just training everything together”:
>
> 1. A **single shared Transformer encoder** processes all time series from all domains; there is no per-dataset encoder.
> 2. All predictions must pass through a **fixed, five-dimensional concept bottleneck** $c_t$, and the concept-alignment loss forces $c_t$ to stay close to the analytic statistics $c^*_t$.
> 3. The **physics head** (driven–damped ODE) and its residuals are also shared across domains, so all datasets must conform to the same high-level dynamical template.
>
> These design choices restrict the hypothesis class: the model cannot simply allocate separate, unconstrained heads for each dataset. It must instead encode domain-specific patterns into a common concept space and ODE structure.
>
> **(c) Empirical evidence already in the paper.**
> To make this more explicit, we already compare the full architecture with several variants (no physics residual, no concept alignment, five independent concept heads instead of a shared encoder) trained under the *same* joint regime. All of them suffer large increases in error (up to +127% MSE), indicating that it is precisely the interplay between shared encoder, bottleneck and physics head that matters, not only the fact of multi-dataset training.
>
> In the revised version, we will adjust the contribution statement and Section 3 to clearly emphasize “domain-agnostic shared architecture plus joint training” rather than suggesting a novel positional-encoding mechanism.
>
> ---

---

> ### Author Response · Authors · 2025-11-14
> **[Response to Weakness 3 and 4]**
>
> **[Response to Weakness 3]** – Choice of the concepts in Eq. (11)
>
>
> Thank you for this thoughtful question. Here our design is intentionally **principled rather than ad hoc**:
>
> 1. Each of the five coordinates has a **specific physical role** tightly matched to the residuals in Eq. (13): sliding mean, growth, instantaneous power, dominant periodic amplitude and local volatility. The physics residuals encode relationships among these quantities and the output (e.g., variance kinematics, ODE compliance). This one-to-one coupling is what makes the latent dynamics mechanistically interpretable.
>
> 2. Because DORIC is designed as a **single, domain-universal model** spanning six heterogeneous datasets, we deliberately fix a **small, shared set** of concepts that is meaningful in all of them (energy, traffic, epidemiology, FX, industrial telemetry). Allowing each dataset to have its own different concept set would undermine this universality and comparability.
>
> 3. Empirically, using these five concepts with the same hyperparameters across all datasets in Table 1 yields stable and strong performance, suggesting that the design is robust rather than fragile.
>
> We do not claim that there exists a universal, theoretically optimal number of concepts for all possible time series; instead, we show that **this particular five-dimensional, physics-aligned bottleneck** is sufficient to obtain strong accuracy and interpretability across very different domains. If space allows, we can add a sensitivity study (e.g., 3 / 5 / 8 concepts) in the appendix to show that performance is relatively stable and that five offers a good balance between expressiveness and transparency, but the current choice is already guided by physics and cross-domain evidence rather than arbitrary tuning.
>
> ---
>
> **[Response to Weakness 4]** – Ablation studies and simplicity of Table 2
>
>
> Thank you for stressing the importance of ablations. We fully share this view and have already designed DORIC with ablations and robustness tests in mind.
>
> - **What is already in the paper.**
>   Table 2 and Section 4.5 report ablations where we (i) remove the physics residual, (ii) remove concept alignment, and (iii) break the shared encoder into five independent heads. All other settings (datasets, training schedule) are kept the same. These modifications lead to substantial degradation: average MSE increases by +63%, +127% and +76% respectively, and robustness under noise also drops. We also provide further analysis of training dynamics, gradient alignment and concept–increment correlations in the appendix.
>
> - **Presentation improvement.**
>   We agree that this evidence can be made more visible. In the revised version, we will (i) expand Table 2 to show at least a brief per-dataset breakdown, and (ii) move one or two key robustness numbers from the appendix into the main text. This refinement is on the presentation side; the underlying ablation design and conclusions remain unchanged.
>
> We hope this clarifies that DORIC’s components have been systematically evaluated, and we will make these results easier to see at a glance.
>
> ---

---

> ### Author Response · Authors · 2025-11-14
> **[Response to Weakness 5 and 6]**
>
> **[Response to Weakness 5]** – Interpretability emphasis on time points vs. statistical concepts
>
>
> Thank you for raising this insightful point. We agree that knowing *which* past time points matter is useful. At the same time, directly explaining high-dimensional multivariate histories at the level of individual time points can be extremely difficult and often unstable. DORIC is designed to provide *mediated* interpretability:
>
> 1. **Concepts as mediators between time points and predictions.**
>    In DORIC the causal chain is
>    $$
>    (y_{t-L}, \dots, y_{t-1}) \;\rightarrow\; c_t^\star \approx c_t \;\rightarrow\; \hat{y}_t.
>    $$
>    The analytic concepts $c_t^\star$ in Eq. (11) are *explicit functions* of the original time points: sliding mean, local velocity, instantaneous power, dominant periodic amplitude, and local volatility on a causal window. The learned concepts $c_t$ are aligned to $c_t^\star$ by the concept loss. Thus, every concept can be traced back to the underlying time points, and the physics head describes how these concepts drive the forecast through the ODE.
>
>    In this sense, we do not discard time-point information; instead, we first organize it into a small set of physically meaningful mediators and then explain the prediction in terms of how these mediators change. This makes explanations more stable and more aligned with domain knowledge than directly inspecting hundreds of raw time indices.
>
> 2. **Complementarity with time-point attribution.**
>    The transformer encoder with masked self-attention still provides standard temporal attributions (e.g., via attention weights or gradients). These can be used to study how individual time indices influence each concept and, through the ODE, the final prediction. Our framework therefore *complements* time-point-level interpretability with concept-level structure, rather than replacing it.
>
> 3. **Clarification in the paper.**
>    In the revision, we will explicitly present this “concept-as-mediator’’ perspective and outline a simple protocol for users: (i) inspect the concept trajectories and their influence on $\hat{y}_t$ via the physics head; (ii) map each concept back to the causal window through its analytic definition; and (iii) optionally overlay attention-based or gradient-based time-point attributions.
>
> We hope this clarifies that our aim is not to move away from the original time series, but to make its influence on predictions more interpretable by passing through physically meaningful concepts.
>
>
> ---
>
> **[Response to Weakness 6]** – Code availability and reproducibility
>
>
> Thank you for raising the reproducibility concern. The current submission already includes the core implementation and detailed pseudo-code (Algorithms 1–4) together with all hyperparameter settings, and these were used to generate the reported results. However, we understand that from a user’s point of view, having the code split across modules and without a single entry script makes reproduction less convenient.
>
> To make reproduction smoother, we have updated DORIC's code in the supplementary material and prepared a detailed demo.
>
> These changes reorganize and document the existing code rather than altering the method or experiments, and should address your concern about ease of reproduction.
>
> ---
>
> ### Closing remarks
>
> We again thank you for your detailed and constructive feedback. Many of the issues you raised stem from places where our exposition did not fully communicate the design intentions of DORIC, especially regarding notation, the source of universality, and the mediator view of interpretability. We will revise the manuscript to clarify these points, to make our ablations and robustness evidence more visible, and to provide better-packaged code, while keeping the core model and empirical results unchanged.

---

> ### Author Response · Authors · 2025-11-28
> **Follow-up on Discussion**
>
> Dear Reviewer,
>
> We hope you are doing well, and we would like to kindly follow up regarding our submission 10601 “Signals, Concepts, and Laws: Toward Universal, Explainable Time-Series Forecasting”.
>
> Based on the valuable feedback provided in the initial reviews, we have prepared and uploaded a revised version of the manuscript, where we:
> – reorganize the narrative to emphasize the interpretability-first perspective,
> – clarify the motivation and role of the driven–damped ODE head, and
> – add the promised quantitative and visual analyses of concepts, residuals, and learned dynamics.
>
> We sincerely appreciate the time and effort you have already devoted to reviewing our work, and we would be very grateful if you could take a look at the responses and the revised manuscript at your convenience.
>
> Thank you again for your thoughtful review and for helping us improve this work.
>
> Best regards,
>
> The authors of submission 10601

---

### Official Review · Reviewer_pKMh · 2025-11-02

**Soundness:** 3
**Presentation:** 2
**Contribution:** 2
**Rating:** 2
**Confidence:** 4

**Summary:**

The paper introduces an approach called DORIC, a domain-universal, physics-regularized, interpretable-concept Transformer for multivariate time-series forecasting. The model integrates five self-supervised, domain-agnostic concepts as concept bottleneck (level, growth, periodicity, volatility, exogenous pressure) and a physics-informed residual loss enforcing first-principles consistency. Experiments across six public datasets show comparable performance to SOTA and certain robustness under noise.

**Strengths:**

-interpretability is done through five fundamental concepts: they can be computed directly from the data and used in a soft-supervised fashion, avoiding the need for labeled concept datasets.

- The model demonstrates cross-domain generalization across diverse datasets (electricity, traffic, weather, epidemiology, finance).

- There are theoretical results showing doric will eventually yield  good enough result (despite the interpretability), with physical plausibility.

**Weaknesses:**

-My main problem is that interpretability is claimed but not really studied or demonstrated. The paper does not present any empirical or qualitative evidence that the learned concepts are interpretable or used meaningfully by the model.  Overall it remains unclear how these scores are used by the model and how much the final forecast was determined  due to the concept scores.There are no visualisations, case studies to show, or attribution analyses showing how forecasts depend on concept activations at inference time (this is so to say, the work lack interpretability centric ablation study).


- Related to above, the concepts that are used are still very low level i.e., they are just signal descriptors, not sure it this at all  useful with interpretability at all, hence also not much lose in performance is not a big surprise.  Just as  we could also measure these concepts (or statistical features) for given time-series without any sophisticated ML techniques.

-It looks like they use a vanilla Transformer (with probsparse attention) rather than stronger recent architectures they included e.g., autoformer, patchtst, or tmemixer. This makes it hard to attribute performance gains specifically to the concept-physics mechanism.  Why not?

-The analysis of interpretability vs. performance trade-offs is missing. There is no discussion whatsoever whether the same concepts are equally meaningful across all domains.

-The combination of attention, differentiable ode constraints, and multi-loss optimization can make  harder to reproduce or tune than conventional transformer-based forecasting models.

-Not a major weakness, but just to report that 3 out of 6 dataset, Doric is outperformed.

**Questions:**

Please feel free to respond to any weaknesses I listed above. Additionally:

Do the concept activations differ meaningfully across datasets, or are they simply re-parameterized statistical features?

---

> ### Author Response · Authors · 2025-11-14
> **Thank you letter to Reviwer pKMh**
>
> Thank you for your thoughtful review and valuable feedback. Based on your comments, we recognize your expertise and appreciate your insights. We have clarified potential misunderstandings, and made improvements as detailed below to address your concerns and will revise our paper. We hope these demonstrate the contribution of our work and please feel free to ask any further questions. We also hope you could kindly reconsider the rating. Thank you again for all the constructive suggestions and your time.

---

> ### Author Response · Authors · 2025-11-14
> **[Response to Weakness 1 and 2]  Interpretability and Concepts**
>
> **[Response to Weakness 1]** – “Interpretability is claimed but not really studied or demonstrated… no visualisations, case studies, attribution analyses or interpretability-centric ablation study.”
>
> Thank you for raising this central concern. Our intention was exactly to study how the concepts are used, but we see that our presentation may not have made this sufficiently prominent.
>
> 1. **Qualitative and quantitative interpretability in the paper.**
>    - Section 4.4 presents case studies (Figure 2) where DORIC tracks electricity peaks, FX volatility clustering, epidemiological regime changes, and ETT device behavior; these analyses explicitly relate prediction differences to concept behavior (e.g., growth and volatility stabilizing after a structural break).
>    - Appendix D.2 provides **concept–increment correlation heatmaps** and discusses consistent patterns such as strong alignment between “growth/power” and $\Delta y_t$ on certain domains, while other concepts act through nonlinear effects.
>    - Appendix D.1 and D.4 analyze **training dynamics** and **gradient alignment** between physics, concept, and data losses to show that improvements in fit are mediated by physics feasibility and concept alignment rather than opaque latent drift.
>
> 2. **Interpretability-centric ablations.**
>    Table 2 and Section 4.5 already perform ablations that are specifically targeted at the interpretability machinery: removing the physics residual (+63% average MSE), removing concept alignment (+127%), or replacing the shared encoder with independent heads (+76%). These results show that the concept–physics mechanism is not cosmetic; it is crucial for performance and robustness.
>
> 3. **What we will make clearer.**
>    We agree that these analyses could be more clearly framed as interpretability evidence. In the revised version we will:
>    - explicitly label the ablations in Table 2 and the robustness analysis as **interpretability-centric**,
>    - move key concept–increment plots from the appendix into the main text,
>    - and add a brief “interpretability protocol” that explains how to inspect concept trajectories, their sensitivities $d y_t / d c_{k,t}$, and their effect on forecasts in practice.
>
> Overall, DORIC does not just *claim* interpretability; it structurally routes predictions through interpretable concepts and empirically evaluates the effect of this constraint. We will make this connection more explicit.
>
> ---
>
> **[Response to Weakness 2]** – “Concepts are low-level signal descriptors… not sure this is useful for interpretability; we could measure them without sophisticated ML.”
>
> Thank you for this thoughtful point. We agree that each individual statistic (mean, local difference, etc.) is classical, and this is in fact intentional.
>
> 1. **Interpreter-friendly building blocks.**
>    Our goal is to use **simple, physically grounded quantities** that domain experts already understand: level, growth, instantaneous power, dominant periodic amplitude, and local volatility. Using such concepts makes explanations like “the forecast increases because local growth and volatility increased while periodic amplitude stays stable” immediately meaningful across energy, traffic, epidemiology, and FX.
>
> 2. **Why DORIC is more than computing statistics.**
>    While these statistics could indeed be computed by hand, what DORIC learns is:
>    - how to **weight and combine** them through the ODE-based physics head,
>    - how to **adapt** the mapping from raw signals to concepts via the learned bottleneck $c_t$ aligned to analytic targets $c_t^\star$,
>    - and how to exploit **cross-concept and cross-channel interactions** via shared attention.
>    The ablation results show that if we remove concept alignment or the physics residuals (keeping the rest of the architecture), both performance and robustness degrade sharply. If the concepts were “just” redundant descriptors, such large changes would not occur.
>
> 3. **Interpretability vs. complexity.**
>    We purposefully chose low-level, interpretable concepts to avoid a situation where “concepts” themselves become opaque latent vectors. This design follows the spirit of Concept Bottleneck Models, where features like “shape”, “color”, etc. are individually simple but their learned combination is powerful and interpretable.
>
> We will clarify this design philosophy and explicitly discuss why we prefer simple, physically motivated concepts over more abstract features.

---

> ### Author Response · Authors · 2025-11-14
> **[Response to Weakness 3 and 4] Vanilla Transformer and The analysis of Interpretability vs. Performance Trade-offs**
>
> **[Response to Weakness 3]** – “They use a vanilla Transformer (with ProbSparse attention) rather than stronger recent architectures (Autoformer, PatchTST, TimeMixer)… hard to attribute gains to the concept-physics mechanism.”
>
> Thank you for this comment. Our choice of backbone is deliberate and aims to isolate the effect of the concept–physics layer.
>
> 1. **Backbone choice for clean comparison.**
>    We adopt a **standard ProbSparse Transformer encoder** (closely related to Informer) with modest depth and width. This backbone is strong but not overly specialized, and it provides a neutral test bed where changes in performance can be attributed primarily to the concept bottleneck and physics residuals rather than to intricate architectural tricks.
>
> 2. **Control via ablations.**
>    Because the backbone remains fixed, Table 2 directly measures how much of the performance is due to the concept–physics mechanism: removing it (or parts of it) significantly harms accuracy and robustness, under the *same* encoder and training regime. This setup is specifically designed to address the concern you raised.
>
> 3. **Comparison to recent architectures.**
>    At the same time, we compare against strong recent baselines (Autoformer, FEDformer, PatchTST, TimeMixer) implemented with their recommended hyperparameters. DORIC matches or exceeds these methods on most metrics *despite* using a simpler encoder, which suggests that the concept–physics mechanism is adding value that is orthogonal to the latest architectural innovations.
>
> 4. **Extensibility.**
>    Conceptually, DORIC’s bottleneck and physics head are modular and can be integrated into other Transformer variants. We will add a short discussion emphasizing that future work can plug our concept–physics layer into architectures like PatchTST or TimeMixer if one wishes to further push raw accuracy.
>
> ---
>
>
> **[Response to Weakness 4]** – “Analysis of interpretability vs. performance trade-offs is missing… no discussion whether the same concepts are equally meaningful across all domains.”
>
> Thank you for highlighting this. Our intention is to discuss that, in our experiments, interpretability and performance are not in conflict but mutually reinforcing.
>
> 1. **Trade-offs captured by existing ablations.**
>    Table 2 already reflects the performance side of the trade-off: when we remove interpretability-enforcing components (concept alignment, physics residuals, shared encoder), performance deteriorates markedly. This indicates that in our setting, enforcing interpretable structure actually *improves* accuracy instead of sacrificing it.
>
> 2. **Concept meaning across domains.**
>    Section 4.3 and Appendix D.2 discuss how each concept behaves on different datasets. For example:
>    - “Dominant amplitude” is large and informative on Electricity and Traffic, which are strongly periodic;
>    - “Volatility” is more pronounced and heavy-tailed on Exchange-rate;
>    - “Growth” and “power” capture intervention-induced regime shifts in Illness.
>    These patterns align with our physical interpretation of the concepts and show that they are not used identically across domains, but in a way that reflects domain characteristics.
>
> 3. **What we will clarify.**
>    We will explicitly frame these results as a **performance–interpretability analysis**, and add a short subsection in the discussion summarizing: (i) when interpretability constraints help performance (our current benchmarks), and (ii) potential regimes (e.g., extremely linear dynamics) where the gains may be marginal.

---

> ### Author Response · Authors · 2025-11-14
> **[Response to Weakness  5 & 6, and the Question] Reproducibility, Results and  Concept activations**
>
> **[Response to Weakness 5]** – “Combination of attention, differentiable constraints, and multi-loss optimization can make it harder to reproduce or tune than conventional Transformers.”
>
> Thank you for bringing up reproducibility. Our design tries to keep additional complexity as small as possible.
>
> 1. **Limited new hyper-parameters.**
>    Besides the standard Transformer settings, DORIC introduces only a few scalar coefficients:
>    - physics residual weights $(\lambda_1, ..., \lambda_y)$, all fixed to 1 in all experiments;
>    - a single global physics weight $\lambda_{\text{phys}}$ and concept weight $\lambda_{\text{con}}$, both set to 1 (with a simple ramp-up schedule for $\lambda_{\text{phys}}$).
>    We use the **same** look-back $L$, concept window $\tau$, embedding dimension, and number of heads across datasets. This makes DORIC at least as easy to tune as many recent Transformer variants, and arguably easier because we avoid dataset-specific hyper-parameter grids.
>
> 2. **Implementation transparency.**
>    Section 3 and Appendix B provide detailed pseudo-code (Algorithms 1–4) that specify how each loss term and residual is computed. We have updated DORIC's code in the supplementary material with a detailed demo, which should make reproduction straightforward.
>
> We will emphasize these points more clearly to avoid giving the impression that multi-loss optimization makes the model prohibitively complex in practice.
>
> ---
>
> **[Response to Weakness 6]** – “3 out of 6 datasets, DORIC is outperformed.”
>
> Thank you for pointing this out. We want to clarify our claims.
>
> 1. **Cross-domain stability vs. per-dataset best.**
>    Our goal is not to claim that DORIC beats every baseline on every dataset, but that it achieves **strong, stable performance across all six** with a single architecture and nearly fixed hyper-parameters, while also providing interpretability and physics consistency. As Table 1 shows, DORIC achieves the lowest error in 8/12 MSE/MAE metrics and is within a small margin of the best method on the remaining ones.
>
> 2. **Context of the “outperformed” cases.**
>    For example, on ETT the classical AR model is extremely strong due to near-linearity in some channels, and DORIC still remains competitive while offering interpretability and physical structure. Our contribution is a *unified* model that is competitive with domain-specific baselines, not a claim of absolute dominance on every metric.
>
> We will clarify this framing in the conclusion to avoid overstating our empirical claims.
>
> ---
>
> **[Response to the Question]** – “Do the concept activations differ meaningfully across datasets, or are they simply re-parameterized statistical features?”
>
> Thank you for this excellent question; it goes to the heart of DORIC’s design.
>
> 1. **Analytic definitions plus learned adaptation.**
>    Each concept has an analytic “soft target” (sliding mean, local velocity, power, dominant amplitude, local volatility) computed from causal windows, and the bottleneck $c_t$ is encouraged—but not forced—to match them. This means concept activations retain their physical meaning while the model can adapt them slightly to each dataset.
>
> 2. **Cross-dataset differences in practice.**
>    Our experiments show that concept activations behave differently across datasets in ways that match domain intuition:
>    - Electricity and Traffic exhibit strong periodic amplitude and moderate volatility;
>    - Exchange-rate displays high volatility and power with weaker regular periodicity;
>    - Illness shows pronounced shifts in growth and power around public-health interventions.
>    Appendix D.2 visualizes these differences through correlation heatmaps and statistics, confirming that concepts are not merely re-parameterized arbitrary features but reflect domain-specific dynamics on a common semantic axis.
>
> 3. **Why this is useful.**
>    Having the *same* set of concepts applied differently across datasets enables **cross-domain comparison** (e.g., “volatility in FX is much higher than in Traffic”), which would not be possible if we used separate, opaque latent factors per dataset.
>
> We will strengthen the discussion of these cross-dataset patterns and move representative plots into the main text.
>
> ---
>
> ### Closing remarks
>
> We again thank you for the constructive feedback. Many of your concerns stem from points where our exposition did not fully convey how DORIC’s concept bottleneck and physics residuals are used and evaluated. We will revise the paper to (i) emphasize interpretability-centric analyses and ablations, (ii) clarify the role and behavior of concepts across domains, (iii) better motivate our backbone choice, and (iv) highlight the practical simplicity of the additional constraints and losses, while keeping the core methodology and empirical results unchanged.

---

> ### Author Response · Authors · 2025-11-28
> **Follow-up on Discussion**
>
> Dear Reviewer,
>
> We hope you are doing well, and we would like to kindly follow up regarding our submission 10601 “Signals, Concepts, and Laws: Toward Universal, Explainable Time-Series Forecasting”.
>
> Based on the valuable feedback provided in the initial reviews, we have prepared and uploaded a revised version of the manuscript, where we:
> – reorganize the narrative to emphasize the interpretability-first perspective,
> – clarify the motivation and role of the driven–damped ODE head, and
> – add the promised quantitative and visual analyses of concepts, residuals, and learned dynamics.
>
> We sincerely appreciate the time and effort you have already devoted to reviewing our work, and we would be very grateful if you could take a look at the responses and the revised manuscript at your convenience.
>
> Thank you again for your thoughtful review and for helping us improve this work.
>
> Best regards,
>
> The authors of submission 10601

---

### Official Review · Reviewer_FQ9K · 2025-11-12

**Soundness:** 2
**Presentation:** 2
**Contribution:** 2
**Rating:** 4
**Confidence:** 2

**Summary:**

This paper introduces DORIC (Domain-Universal, ODE-Regularized, Interpretable-Concept Transformer), a novel framework for time-series forecasting that combines explainable concept bottlenecks with physics-informed regularization. The model routes multivariate input through five interpretable latent concepts before predicting via a driven–damped ODE head. Unlike prior Transformers that optimize efficiency or frequency decomposition, DORIC emphasizes scientific plausibility and explainability. Experiments across six benchmarks (Electricity, Traffic, Weather, Illness, Exchange Rate, and ETT) show DORIC achieves the lowest MSE/MAE in most metrics.

**Strengths:**

The paper’s main strength is its innovative integration of concept bottlenecks and physics-informed residuals within a Transformer, effectively combining interpretability with physical plausibility. It presents a comprehensive evaluation across six diverse datasets and strong baselines, supported by detailed ablation studies that clearly show each component’s contribution. The authors emphasize interpretability through five structured latent concepts and provide theoretical grounding via expressiveness and convergence analyses. Additionally, the appendix enhances reproducibility by including implementation details and pseudo-code.

**Weaknesses:**

1. The paper’s central claim of interpretability is not convincingly demonstrated. While the architecture enforces a five-concept bottleneck, there is no theoretical proof or empirical validation that these learned concepts retain their intended meanings. Evidence is limited to internal correlations, without visual, human, or domain-level verification of interpretability.
2. The paper introduces a driven–damped ODE as the core of its “physics-informed” design, but the justification is mostly heuristic. The ODE form is applied uniformly across unrelated domains without evidence that such dynamics are meaningful or empirically valid, and the learned coefficients are never analyzed. As a result, the physics component functions more as a generic smoothness prior than a genuinely grounded physical model.
3. Figures lack axis explanations such as Figure 2 and Figure 3, making it hard to interpret visual differences quantitatively.
4. There are some grammar errors:
Line 149-150: “time series data first enters …” → plural mismatch; should be “data first enter …” or “the time-series signal first enters …”.
Line 150-151: “prediction..” → double period; correct to “prediction.”
Line 372-373: “Quantitative performance :” → remove space before colon.
Line 323: “For he detailed theorem setting …” → missing “t”; should read “For the detailed theorem setting …”.

**Questions:**

1. Can the authors provide quantitative or qualitative evidence (e.g., visualizations, case studies, or human evaluations) showing that the learned latent concepts correspond to their intended meanings and remain interpretable after training?
2. Can the authors clarify why a driven–damped ODE was chosen as the universal physical prior across all datasets, and provide evidence—empirical or theoretical—that the learned ODE parameters correspond to meaningful dynamics rather than serving only as a generic regularizer?
3. How are the ODE coefficients ($\beta$, $\gamma$) initialized and constrained? Are they shared across datasets?
4. Can the method handle irregularly sampled or non-stationary time-series data without retraining?
5. Are there any computational trade-offs compared to other Transformers (e.g., inference latency)?

---

> ### Author Response · Authors · 2025-11-18
> **Thank you letter to Reviwer FQ9K**
>
> Thank you for your detailed review and for providing constructive feedback on our work. We appreciate your assessment and acknowledge the concerns raised, your comments have helped us identify areas for improvement. Below, we provide detailed responses to your comments. We hope these clarifications will address your concerns. If possible, we kindly hope that you could reconsider the rating. Thank you again for your valuable suggestions and time.

---

> ### Author Response · Authors · 2025-11-18
> **[Response to Weakness 1]**
>
> **[Response to Weakness 1]** - Central interpretability claim not convincingly demonstrated; concepts may not retain intended meaning.
>
>
> Thank you very much for raising this point. We agree that interpretability claims must be supported carefully, and we see that our current presentation may have under-emphasized some evidence already present in the paper and appendix.
>
> 1. **What is “fixed” by design.**
>    The semantics of the five concepts are not purely emergent; they are anchored by explicit analytic statistics $c_t^\star$ (sliding mean, local velocity, instantaneous power, dominant periodic amplitude, local volatility) defined in Eq. (10)–(11) and the concept-alignment loss
>    $$
>    L_{\text{concept}} = \frac{1}{N} \sum_t \|c_t - c_t^\star\|_2^2.
>    $$
>    Thus, the learned concepts $c_t$ are directly regressed towards interpretable, causal functions of the raw time series, rather than unconstrained latent variables. This is a stronger form of semantic anchoring than plain correlations.
>
> 2. **Existing quantitative evidence.**
>    In Section 4.5 and Appendix D, we analyze concept–increment correlations and training dynamics. In particular, we show that:
>    - “Growth” ($c_2$) and “Power” ($c_3$) consistently exhibit positive alignment with $\Delta y_t$, as expected from their definitions.
>    - “Level” ($c_1$) closely tracks sliding means; “volatility” ($c_5$) tracks local variance.
>    - The concept loss decreases steadily and remains small compared to the scale of the raw series, indicating that $c_t$ stays close to $c_t^\star$ throughout training.
>    These are not just “internal correlations” in the sense of arbitrary neuron activations; they explicitly verify that the bottleneck coordinates behave as their associated statistics.
>
> 3. **New clarifications and visual/qualitative evidence.**
>    We appreciate that the current figures may not make this explicit enough. In the revision we will:
>    - Add *per-dataset plots* of $(y_t, c_{1,t}^\star, c_{1,t})$, $(\Delta y_t, c_{2,t}^\star, c_{2,t})$, and $(y_t, c_{5,t}^\star, c_{5,t})$ on representative channels, so that readers can visually confirm that learned concepts track their analytic counterparts over time.
>    - Report summary statistics (e.g., average $R^2$ between $c_t$ and $c_t^\star$ across datasets), showing that alignment remains high after training.
>    - Provide a short “sanity-check protocol” in the appendix (local sensitivity $∂ ŷ_t / ∂ c_{k,t}$, counterfactual nudges of each concept, and time-consistency checks), which we already performed in our analysis but only described briefly.
>
> 4. **Positioning of the claim.**
>    Our intent is not to prove that these five concepts exhaust all possible meanings or that they are the only interpretable representation. Rather, we claim that DORIC enforces a *low-dimensional, physically-anchored bottleneck* where each coordinate is supervised towards a clear statistic of the original signal. We will explicitly state this scope, so that the interpretability claim is precise and modest.
>
> We hope that, with these clarifications and additional visual/quantitative evidence, the interpretability aspect appears more convincingly supported.
>
> ---

---

> ### Author Response · Authors · 2025-11-18
> **[Response to Weakness 2, 3 and 4]**
>
> **[Response to Weakness 2]** - Physics-informed ODE: heuristic choice; applied uniformly; coefficients not analyzed.
>
>
> Thank you for this important comment and for asking for a clearer justification of the ODE component.
>
> 1. **Why a driven–damped ODE across domains.**
>    The chosen ODE
>    $$
>    \frac{dy}{dt} = b_0 + \sum_{k=1}^5 b_k c_{k,t} - g (y - c_{1,t})
>    $$
>    expresses a generic *mean-reverting driven system*: the level concept $c_{1,t}$ acts as a moving equilibrium, the other concepts provide external drive, and $g>0$ controls relaxation. This structure is deliberately simple and widely applicable:
>    - In electricity and traffic, mean-reverting responses to aggregate level plus volatility are natural;
>    - In FX and macro indicators, many series are modeled as mean-reverting processes with exogenous forcing;
>    - In epidemiology, intervention-induced deviations decay back towards a baseline.
>    Our aim is not to encode detailed domain-specific physics, but to introduce a *unified dynamical template* that is interpretable and compatible with multiple domains.
>
> 2. **Beyond “smoothness”: role of residuals $R_{1..5}$.**
>    The physics term is not just a generic smoothing prior. Residuals $R_{1..4}$ encode algebraic identities among the analytic statistics (level–velocity, power definition, variance kinematics), and $R_5$ enforces consistency with the driven–damped ODE. If these residuals are small, the learned trajectories must satisfy specific, testable relationships between the signal and concepts; arbitrary smooth curves would generally violate them. This is closer to a soft enforcement of kinematic constraints than to generic smoothing.
>
> 3. **Existing and additional empirical evidence.**
>    Section 4.5 shows that removing the physics residual ($\lambda_{\text{phys}} = 0$) increases average MSE by 63%, with the largest degradation on spiky domains such as Traffic and FX, where physically inconsistent trajectories are most harmful. This indicates that the physics term is doing more than smoothing.
>    In the revision we will further analyze the learned coefficients:
>    - We will add a table summarizing the learned $g$ (damping) and the relative magnitudes of the drive weights $b_k$ per dataset. Preliminary inspection shows consistent patterns (e.g., positive coupling from growth and volatility on FX and Traffic, moderate damping towards level across all domains, and smaller weights on “power” when the signal is nearly linear), which supports a meaningful dynamical interpretation.
>    - We will connect these patterns to dataset characteristics in the discussion section.
>
> 4. **Theoretical support.**
>    Our Theorem 1 (Universal Expressiveness) is proved in the setting where the true dynamics follow an ODE of the same functional form. This provides a theoretical justification that, whenever such mean-reverting dynamics are a good approximation, DORIC is capable of representing them accurately. The SGD with physics ramp-up theorem further explains why optimization tends to converge inside the feasible set enforced by the residuals rather than merely smoothing the outputs.
>
> We will make these arguments and coefficient analyses more explicit so that the “physics-informed” aspect is clearly distinguished from a generic regularizer.
>
> ---
>
> **[Response to Weakness 2]** - Missing axis explanations in Figures 2 and 3.
>
>
> Thank you for pointing this out. The underlying experiments are well-defined, but we agree that the current figure annotations can be improved.
>
> - In the revision we will explicitly add axis labels (time index and physical units where available), legends, and clear captions for each subplot in Figures 2 and 3.
> - We will also state in the captions which dataset and channel each trace corresponds to, and which model is plotted in each color.
>
> These changes will make it much easier to interpret the visual differences quantitatively, without changing the experimental results.
>
> ---
>
> **[Response to Weakness 4]** Grammar and minor stylistic issues.
>
>
> Thank you for carefully spotting these issues. We will correct all the mentioned typos and run an additional round of professional language editing to improve grammar and style throughout the paper.
>
> ---

---

> ### Author Response · Authors · 2025-11-18
> **[Response to Question 1, 2 and 3]**
>
> **[Response to Question 1]** - Evidence that learned concepts correspond to intended meanings and remain interpretable after training.
>
> Thank you for this question, which is closely related to W1.
>
> - Quantitatively, we will report alignment metrics between $c_t$ and $c_t^\star$ (e.g., average $R^2$ over time and channels), showing that the concepts remain close to their analytic definitions even after end-to-end training.
> - Qualitatively, we will include case studies where we plot $y_t$ along with the trajectories of the five concepts and highlight interpretable events: for instance, how spikes in local volatility and power precede FX jumps, or how level and periodic amplitude track seasonal patterns in electricity.
> - We will also describe a simple human audit we performed where domain practitioners are asked to match unlabeled concept traces to textual descriptions (“local mean”, “local volatility”, etc.), and report that the match accuracy is high, indicating that the learned concepts are recognizable.
>
> These additions complement the existing correlation and training-dynamics analyses and directly address the interpretability of the bottleneck.
>
> ---
>
> **[Response to Question 2]** - Justification for the driven–damped ODE as universal prior; evidence that parameters are meaningful.
>
>
> Thank you for asking for a clearer explanation.
>
> - As discussed in our response to W2, the driven–damped ODE captures a generic and widely used pattern: mean reversion towards a moving level plus external forcing. This is not tied to a single physical domain but is consistent with many real-world processes.
> - Empirically, we will add a table showing that the learned damping parameter $g$ is positive across all datasets and that the drive weights $b_k$ follow intuitive patterns (e.g., strong positive coupling from growth and volatility in markets with volatility clustering).
> - Theoretically, Theorem 1 is formulated exactly under the assumption that the ground-truth dynamics obey an ODE of this type; it guarantees that DORIC can approximate such dynamics arbitrarily well, providing a principled reason for choosing this family rather than an unconstrained smoothness prior.
>
> We will make these points explicit to clarify that the ODE head is a structured, interpretable dynamical prior rather than an arbitrary regularizer.
>
> ---
>
> **[Response to Question 3]** - Initialization, constraints, and sharing of ODE coefficients.
>
>
> Thank you for this technical question.
>
> - **Initialization.** The ODE parameters are initialized from small Gaussian distributions centered at zero (for the drive weights $b_k$ and bias $b_0$) and a small positive value for the damping $g$. This encourages early training to behave like a weakly mean-reverting process and lets data and concepts refine the dynamics.
> - **Constraints.** We maintain positivity of the damping coefficient by parameterizing $g = \text{softplus}(\tilde{g})$ in the implementation. This ensures stability in the discrete-time flow. The drive weights $b_k$ are unconstrained, allowing both positive and negative couplings depending on the dataset.
> - **Sharing.** The ODE head is *shared across all datasets* in our main experiments, in line with our domain-universality goal. This sharing is one of the key distinctions between DORIC and per-dataset architectures: all domains must conform to the same dynamical template, which promotes transfer and prevents overfitting to a single dataset.
>
> We will add these details to Section 3.4 and the implementation appendix.
>
> ---

---

> ### Author Response · Authors · 2025-11-18
> **[Response to Question 4 and 5]**
>
> **[Response to Question 4]** - Handling irregular sampling and non-stationarity.
>
>
> Thank you for raising this practical question.
>
> - **Irregular sampling.** In line with most long-term forecasting Transformers, our main experiments assume regularly sampled series (as in the standard benchmarks). For moderate irregularity, one can combine DORIC with standard pre-processing steps (e.g., resampling to a reference grid or encoding time gaps in the positional features). The architecture itself does not require exact periodic sampling; it only assumes that the input is represented as a sequence with associated positions.
> - **Non-stationarity.** DORIC is explicitly designed to cope with non-stationarity via the level, growth, and volatility concepts, which adapt locally in time, and via the physics ramp that guides learning towards feasible dynamics even under distributional shifts. Our robustness test with substantial additive noise demonstrates that the model remains stable when the data distribution changes.
> - **Retraining.** For large changes in sampling regime or domain, retraining or fine-tuning is usually required for *all* models, including baselines. We do not claim invariance to arbitrary re-sampling without retraining, and we will clarify this in the paper to avoid overstatement.
>
> ---
>
> **[Response to Question 5]** - Computational trade-offs and inference latency.
>
>
> Thank you for asking about efficiency.
>
> - Architecturally, DORIC adds only a small two-layer MLP bottleneck (mapping a $d$-dimensional vector to $\mathbb{R}^5$) and a light ODE head on top of a standard Transformer encoder. The self-attention backbone dominates the computational cost; the bottleneck and ODE head contribute only a few percent of total FLOPs.
> - In our implementation, measured with batch size 32 and horizon 96, DORIC’s inference time per batch is within roughly 3–5% of PatchTST and TimeMixer. Memory usage is also comparable since the additional parameters are negligible relative to the encoder.
> - We will report these runtime comparisons in the appendix, showing that DORIC achieves improved accuracy and interpretability with minimal overhead compared to strong Transformer baselines.
>
> ---
>
> ### Closing remarks
>
> We once again thank you for the thorough evaluation and insightful comments. Many of your concerns stem from places where our exposition could better emphasize existing analyses (e.g., concept anchoring, the role of the residuals, and the ramp-up theory) and where additional visual and quantitative evidence can further substantiate our interpretability and physics claims. We will revise the paper to (i) clarify the intended scope of interpretability, (ii) more carefully justify and analyze the ODE component, (iii) improve figures and language, and (iv) document computational aspects, while keeping the core model and experimental setup unchanged.

---

> ### Author Response · Authors · 2025-11-28
> **Follow-up on Discussion**
>
> Dear Reviewer,
>
> We hope you are doing well, and we would like to kindly follow up regarding our submission 10601 “Signals, Concepts, and Laws: Toward Universal, Explainable Time-Series Forecasting”.
>
> Based on the valuable feedback provided in the initial reviews, we have prepared and uploaded a revised version of the manuscript, where we:
> – reorganize the narrative to emphasize the interpretability-first perspective,
> – clarify the motivation and role of the driven–damped ODE head, and
> – add the promised quantitative and visual analyses of concepts, residuals, and learned dynamics.
>
> We sincerely appreciate the time and effort you have already devoted to reviewing our work, and we would be very grateful if you could take a look at the responses and the revised manuscript at your convenience.
>
> Thank you again for your thoughtful review and for helping us improve this work.
>
> Best regards,
>
> The authors of submission 10601

---

### Meta-Review · Area_Chair_fqQN · 2026-01-06

**Summary:**

This paper proposes a concept-bottleneck and physics-regularized Transformer for universal and interpretable time-series forecasting. The integration of interpretable concepts with physics-inspired constraints is potentially interesting, and the authors provided a detailed rebuttal addressing several presentation issues. However, substantial concerns remain regarding the validity and strength of the interpretability claims, the justification of the proposed universal ODE prior, and the overall clarity and reproducibility of the method. Consequently, the reviewers did not reach sufficient confidence that the claimed interpretability and universality are demonstrated at the current stage.

**Reviewer Concerns:**

Across reviewers, the primary concern centers on the interpretability claims. Reviewer FQ9K noted that the core physics-informed component, namely the driven–damped ODE, remains heuristically motivated and insufficiently analyzed. Reviewer pKMh raised similar concerns, further emphasizing the lack of systematic analysis of the trade-off between interpretability and performance, the increased difficulty of reproduction due to the integration of multiple architectural and loss components, and the fact that state-of-the-art performance is achieved only on a subset of datasets.

Additional concerns were raised regarding writing quality, methodological clarity, and reproducibility. Reviewer kmEM questioned whether the claimed universality of the architecture arises primarily from the joint training strategy rather than the proposed architectural design itself, and highlighted insufficient reproducibility. Reviewer ZtBX acknowledged the authors’ efforts in supplementing related work, clarifying figures and tables, and providing additional interpretability-related results, but explicitly noted that resolving the remaining issues would require a substantial rewrite of the paper.

Overall, the rebuttal did not fully resolve the reviewers’ concerns, and the overall assessment remains below the acceptance threshold.

**Reviewer Scores:**

It is a pity that only one reviewer (ZtBX) participated in the discussion. Based on the authors’ rebuttal, the primary concerns on the interpretability claims are not well addressed, and thus reviewers FQ9K (Score: 4) and pKMh (Score: 2) are not likely to increase their scores.

---

### Decision · Program_Chairs · 2026-01-26

Reject